# Aberrant epigenome in iPSC-derived dopaminergic neurons from Parkinson's disease patients

Rubén Fernández-Santiago[1,2,3,*], Iria Carballo-Carbajal[2,4,†], Giancarlo Castellano[5,†], Roger Torrent[6], Yvonne Richaud[7,8], Adriana Sánchez-Danés[6], Roser Vilarrasa-Blasi[5], Alex Sánchez-Pla[9,10], José Luis Mosquera[9], Jordi Soriano[11], José López-Barneo[2,12], Josep M Canals[2,3,13], Jordi Alberch[2,3,13], Ángel Raya[7,8,14], Miquel Vila[2,4,14], Antonella Consiglio[6,15,‡], José I Martín-Subero[5,‡], Mario Ezquerra[1,2,3,‡,**] & Eduardo Tolosa[1,2,3,16,‡]

## Abstract

The epigenomic landscape of Parkinson's disease (PD) remains unknown. We performed a genomewide DNA methylation and a transcriptome studies in induced pluripotent stem cell (iPSC)-derived dopaminergic neurons (DAn) generated by cell reprogramming of somatic skin cells from patients with monogenic LRRK2-associated PD (L2PD) or sporadic PD (sPD), and healthy subjects. We observed extensive DNA methylation changes in PD DAn, and of RNA expression, which were common in L2PD and sPD. No significant methylation differences were present in parental skin cells, undifferentiated iPSCs nor iPSC-derived neural cultures not-enriched-in-DAn. These findings suggest the presence of molecular defects in PD somatic cells which manifest only upon differentiation into the DAn cells targeted in PD. The methylation profile from PD DAn, but not from controls, resembled that of neural cultures not-enriched-in-DAn indicating a failure to fully acquire the epigenetic identity own to healthy DAn in PD. The PD-associated hypermethylation was prominent in gene regulatory regions such as enhancers and was related to the RNA and/or protein downregulation of a network of transcription factors relevant to PD (FOXA1, NR3C1, HNF4A, and FOSL2). Using a patient-specific iPSC-based DAn model, our study provides the first evidence that epigenetic deregulation is associated with monogenic and sporadic PD.

**Keywords** DNA methylation; dopaminergic neuron; induced pluripotent stem cell; Parkinson's disease; transcription factor
**Subject Categories** Chromatin, Epigenetics, Genomics & Functional Genomics; Neuroscience; Stem Cells

## Introduction

Parkinson's disease (PD) is a neurodegenerative disorder associated with the progressive loss of dopaminergic neurons (DAn) in the

1 Laboratory of Neurodegenerative Disorders, Department of Neurology, Hospital Clínic of Barcelona, Institut d'Investigacions Biomèdiques August Pi i Sunyer (IDIBAPS), University of Barcelona (UB), Barcelona, Spain
2 Centro de Investigación Biomédica en Red de Enfermedades Neurodegenerativas (CIBERNED), Madrid, Spain
3 Cell Therapy Program, Faculty of Medicine, University of Barcelona (UB), Barcelona, Spain
4 Neurodegenerative Diseases Research Laboratory, Hospital Vall d'Hebron, Vall d'Hebron Research Institute (VHIR), Universitat Autònoma de Barcelona (UAB), Barcelona, Spain
5 Department of Pathological Anatomy, Pharmacology and Microbiology, University of Barcelona (UB), Institut d'investigacions Biomèdiques August Pi i Sunyer (IDIBAPS), Barcelona, Spain
6 Institute for Biomedicine (IBUB), University of Barcelona (UB), Barcelona, Spain
7 Control of Stem Cell Potency Group, Institute for Bioengineering of Catalonia (IBEC), Barcelona, Spain
8 Centre for Networked Biomedical Research on Bioengineering, Biomaterials and Nanomedicine (CIBER-BBN), Zaragoza, Spain
9 Department of Statistics, University of Barcelona (UB), Barcelona, Spain
10 Department of Statistics, Vall d'Hebron Research Institute (VHIR), Barcelona, Spain
11 Departament d'Estructura i Constituents de la Matèria (ECM), Facultat de Física, University of Barcelona (UB), Barcelona, Spain
12 Institute of Biomedicine of Seville (IBiS), Hospital Universitario Virgen del Rocío, Consejo Superior de Investigaciones Científicas (CSIC), University of Seville, Seville, Spain
13 Department of Cell Biology, Immunology and Neuroscience, Faculty of Medicine, Institut d'Investigacions Biomèdiques August Pi i Sunyer (IDIBAPS), University of Barcelona (UB), Barcelona, Spain
14 Institució Catalana de Recerca i Estudis Avançats (ICREA), Barcelona, Spain
15 Department of Molecular and Translational Medicine, University of Brescia and National Institute of Neuroscience, Brescia, Italy
16 Movement Disorders Unit, Department of Neurology, Hospital Clínic of Barcelona, Institut d'Investigacions Biomèdiques August Pi i Sunyer (IDIBAPS), University of Barcelona (UB), Barcelona, Spain
*Corresponding author. Tel: +34 932 275 400 ext. 4814; Fax: +34 935 275 783; E-mail: ruben.fernandez.santiago@gmail.com
**Corresponding author. Tel: +34 932 275 400 ext. 4814; Fax: +34 935 275 783; E-mail: ezquerra@clinic.ub.es
†These authors contributed equally to this work
‡Co-senior authors

substantia nigra pars compacta (SNpc) (Lang & Lozano, 1998a,b). Yet PD is recognized as a systemic disease affecting other tissues apart from the nervous system (Hoepken *et al*, 2008; Beach *et al*, 2010; Shannon *et al*, 2012). Although most cases are sporadic, around 5–10% encompass monogenic forms caused by pathogenic mutations in PD-associated genes (Farrer, 2006). Among these, missense mutations in the leucine-rich repeat kinase 2 (*LRRK2*) gene are the most frequent cause of familial PD (Paisan-Ruiz *et al*, 2004; Zimprich *et al*, 2004) and also of the common sporadic form. The *LRRK2* G2019S mutation alone explains up to 6% familial and 3% sporadic PD cases in Europeans (Di Fonzo *et al*, 2005; Gilks *et al*, 2005) and up to 20% of total cases among Ashkenazy Jews (Ozelius *et al*, 2006) or 40% in North African Berbers (Lesage *et al*, 2006). In addition, *LRRK2*-associated PD (L2PD) is clinical and neuropathologically similar to sporadic PD (sPD) lacking *LRRK2* mutations (Healy *et al*, 2008), thus representing a valuable system to investigate the most common form of disease. Moreover, the reduced penetrance of G2019S in L2PD suggests the involvement, akin to sPD, of yet unknown disease-modifying factors (Healy *et al*, 2008).

Genetic and epigenetic alterations contribute to the physiopathology of diseases (Bergman & Cedar, 2013). In PD, beyond largely studied genetic defects, the epigenomic landscape of disease remains unknown (Urdinguio *et al*, 2009; van Heesbeen *et al*, 2013). Epigenetic modifications are inheritable changes of gene expression without alterations in the DNA sequence which can virtually capture the influence of environmental factors (Feil & Fraga, 2011). Thus, epigenetic alterations can reflect the relationship among factors which have been postulated to play a role in complex neurodegenerative disorders such as the individual genetic background, the environment, and the aging process. To date, epigenetic studies in central nervous system disorders have been hampered by the inaccessibility to disease targeted cells from patients, and especially of DAn from the SNpc in PD. The few published reports in PD were performed in blood cells using single gene candidate approaches (Kontopoulos *et al*, 2006; Pieper *et al*, 2008; de Boni *et al*, 2011; Jin *et al*, 2014) or human postmortem cortex and cerebellum representing end-points of disease (Desplats *et al*, 2011; IPDGC, 2011; Masliah *et al*, 2013). Results from these studies have been sometimes conflicting and did not yield robust associations. In addition, previous studies on iPSC-derived DAn did not explore in the role of the epigenome in PD (Byers *et al*, 2011; Nguyen *et al*, 2011; Seibler *et al*, 2011; Cooper *et al*, 2012; Jiang *et al*, 2012; Sanchez-Danes *et al*, 2012b; Rakovic *et al*, 2013; Reinhardt *et al*, 2013; Ryan *et al*, 2013; Schondorf *et al*, 2014).

In the present study, we investigated the epigenome of PD by performing a genomewide DNA methylation study of CpG dinucleotides and a transcriptome study using an *in vitro* PD model of patient-specific disease-relevant cells (DAn). This cell system consisted in induced pluripotent stem cell (iPSC)-derived DAn generated upon cell reprogramming of parental skin cells from L2PD patients carrying the G2019S mutation (*n* = 4), sPD patients without *LRRK2* mutations (*n* = 6), and gender- and age-matched healthy subjects (*n* = 4) (Sanchez-Danes *et al*, 2012a,b). Studied cell lines were similar in PD and controls as regards their properties and their maturation state and included 30-day morphologically and functionally mature ventromedial (vm)-DAn which were mostly of the A9 subtype (Sanchez-Danes *et al*, 2012b). The goal of our study was to explore for the first time the epigenomic landscape of PD using iPSC-derived DAn, and to compare the methylome of the common sPD form with the uniquely resembling L2PD monogenic form.

## Results

Studied cell lines from PD patients and healthy controls were generated and characterized in parallel blind to researcher in a previous study (Sanchez-Danes *et al*, 2012b) using a 30-days differentiation protocol (Sanchez-Danes *et al*, 2012a) (Table 1, Fig 1 and Materials and Methods). Resulting iPSC-derived DAn had similar morphological and functional properties as well as similar full DAn maturation state in PD and controls (Fig 1) (Sanchez-Danes *et al*, 2012b). Yet consistently with the late onset of disease, the DAn cells from PD patients developed specific neurodegenerative phenotypes upon long-term culture (75-days) including impaired axonal outgrowth, deficient autophagic vacuole clearance, and accumulation of α-synuclein (SNCA) (Sanchez-Danes *et al*, 2012b; Orenstein *et al*, 2013).

### Epigenetic changes are associated with monogenic and sporadic PD

We performed a comprehensive genomewide DNA methylation analysis of 30-days iPSC-derived DAn using the Illumina 450k methylation platform (Bibikova *et al*, 2011). Unsupervised hierarchical clustering of CpGs methylation values showed different DNA methylation profiles between both forms of PD (L2PD and sPD) and controls indicating robust differences between PD and controls (Figs 2A and EV1A). We further detected 1,261 differentially methylated CpG sites (DMCpGs) in L2PD and 2,512 in sPD with respect to controls under an absolute mean methylation difference above 0.25 (Bibikova *et al*, 2011) and an adjusted *P* below 0.05 (Figs 2B and EV1B, and Table EV1). Most DMCpGs in L2PD were common to sPD (78%) and no significant methylation differences were found when comparing L2PD and sPD using the same criteria mentioned above, indicating that L2PD and sPD share similar methylation profiles. Accordingly, both groups were merged for further analysis. In all PD subjects, we identified 2,087 DMCpGs as compared to controls including hypermethylation in 1,046 regions and hypomethylation in 1,041. DMCpGs mostly affected gene bodies and promoters but were also enriched at intergenic regions. Hypermethylated DMCpGs were more often located outside CpG islands, shores, or shelves (73% vs. 31% in background, $P < 0.4 \times 10^{-14}$) (Fig 2C). Notably, genes associated with DMCpGs were largely involved in neural functions and transcription factor (TF) activity (Table EV2). These data indicate that DAn from PD patients show epigenetic abnormalities. They also indicate that monogenic L2PD and the sporadic form of PD share similar DNA methylation changes.

### Epigenetic changes manifest only in DAn from PD patients

We further investigated whether those DNA methylation changes observed in DAn were already present in fibroblasts or in undifferentiated iPSCs from the same subjects by analyzing a subset of representative individuals (two L2PD, two sPD, and three controls). Isogenic fibroblasts and iPSCs from these subjects showed no methylation differences between PD and controls neither for the

**Table 1.  Summary of clinical features and iPSC-derived DAn cell line details from PD patients and gender- and age-matched healthy controls.**

| Cell line code | Code previous study[Ref.] | Subject type | *LRRK2* mutation | Family history of PD | Gender | Age at donation | Age at onset | Initial symptoms[a] | L-DOPA response | Code of selected iPSCs clones | Cell ratio TUJ1+/DAPI+ (neurons)[b] | Cell ratio TH+/TUJ1+ (DA neurons)[c] |
|---|---|---|---|---|---|---|---|---|---|---|---|---|
| C-01 | SP-15 | Control | No | No | Female | 47 | – | – | – | 15-2 | 34.7 | 45.0 |
| C-02 | SP-11 | Control | No | No | Female | 48 | – | – | – | 11-1 | 40.0 | 59.9 |
| C-03 | SP-09 | Control | No | No | Male | 66 | – | – | – | 9-4 | 52.2 | 55.5 |
| C-04 | SP-17 | Control | No | No | Male | 52 | – | – | – | 17-2 | 54.0 | 65.8 |
| PD-01 | SP-13 | L2PD | G2019S | Yes | Female | 68 | 57 | T | Good | 13-4 | 47.0 | 65.2 |
| PD-02 | SP-02 | sPD | No | No | Male | 55 | 48 | T | N/A | 2-1 | 20.6 | 42.2 |
| PD-03 | SP-05 | L2PD | G2019S | Yes | Male | 66 | 52 | B | Good | 5-1 | 39.9 | 49.6 |
| PD-04 | SP-16 | sPD | No | No | Female | 51 | 48 | B | N/A | 16-2 | 32.1 | 55.2 |
| PD-05 | SP-06 | L2PD | G2019S | Yes | Male | 44 | 33 | T | Good | 6-2 | 40.9 | 61.9 |
| PD-06 | SP-10 | sPD | No | No | Male | 58 | 50 | D | Good | 10-2 | 35.0 | 41.1 |
| PD-07 | SP-12 | L2PD | G2019S | Yes | Female | 63 | 49 | T | Good | 12-3 | 42.7 | 60.0 |
| PD-08 | SP-04 | sPD | No | No | Male | 46 | 40 | B | Good | 4-2 | 52.1 | 47.3 |
| PD-09 | SP-01 | sPD | No | No | Female | 63 | 58 | T and B | N/A | 1-1 | 32.2 | 44.9 |
| PD-10 | SP-08 | sPD | No | No | Female | 66 | 60 | T | Good | 8-1 | 41.6 | 67.1 |

N/A, not assessed; C, control.

[a]Initial symptoms; T, tremor; B, bradykinesia; D, foot dystonia.

[b]Ratio of neurons/total cells, estimated by immunofluorescence as the ratio of TUJ1 (neuron-specific class III b-tubulin)-positive cells/DAPI-positive cells.

[c]Ratio of iPSC-derived DAn/total neurons, estimated by immunofluorescence as the ratio of TH (tyrosine hydroxylase)-positive cells/TUJ1-positive cells.

[b] and [c] were calculated blind to researcher upon three independent differentiations as previously described (Sanchez-Danes et al, 2012b).

2,087 DMCpGs detected in iPSC-derived DAn (Fig 2D) nor for any other given CpG (Fig EV1C) under an absolute mean methylation difference above 0.25 (Bibikova et al, 2011) and an adjusted *P* below 0.05. Yet fibroblasts, undifferentiated iPSCs and iPSC-derived DAn showed distinct DNA methylomes as expected for each specific cell type (Doi et al, 2009; Hochedlinger & Plath, 2009) (Fig EV1D). Interestingly, a total of 75% of all the 2,087 DMCpGs did not change in the differentiation from iPSCs to DAn in PD patients, whereas only 40% DMCpGs remained unchanged in controls, suggesting an incomplete epigenomic remodeling in PD in spite of the successful reprogramming and differentiation into mature DAn (Sanchez-Danes et al, 2012b) (Table EV3 and Fig EV2). These findings suggest that latent molecular defects in skin cells from PD patients become uncovered upon differentiation into DAn. This is compliant with the idea of PD as a systemic disease affecting other tissues apart from the nervous system (Hoepken et al, 2008; Beach et al, 2010; Shannon et al, 2012) including molecular defects which were previously reported in fibroblasts from L2PD or sPD patients (Papkovskaia et al, 2012; Ambrosi et al, 2014; Yakhine-Diop et al, 2014). In addition, to address the extent of the methylation changes attributed to DAn alone in the context of cell heterogeneity inherent to iPSC-derived DAn systems, we differentiated iPSCs from the same subjects into neural cultures not-enriched-in-DAn by omitting the lentiviral-mediated forced expression of the ventral midbrain determinant LMX1A and the supplementation with DAn patterning factors. This protocol results in a > 4-fold depletion in the number of DAn (See Materials and Methods and (Sanchez-Danes et al, 2012a)). In neural cultures not-enriched-in-DAn, we did not find methylation differences between PD and controls under an absolute mean methylation difference above 0.25 (Bibikova et al, 2011) and an adjusted *P* below 0.05 (Fig EV1C). Moreover, the methylation profile from PD DAn was closer to that from neural cultures not-enriched-in-DAn as compared to control DAn (Fig 2D). For any

given comparison and using the same restrictive cutoffs mentioned above, we found that the overall methylation variability referred to all samples was attributable, in decreasing order, to (i) the different cell types as expected, (ii) the condition health/disease only in iPSC-derived DAn, and (iii) inter-individual differences in a relative lesser extent. Altogether, these results indicate that the identified PD epigenetic changes are specific for DAn cells and consist in the failure of PD DAn to fully acquire the mature epigenetic identity own to healthy DAn.

**Gene and protein expression changes occur along with methylation changes**

We next studied the transcriptome of iPSC-derived DAn by using gene expression microarrays. Compliant to the methylome analysis, no differentially expressed gene (DEG) was found between L2PD and sPD, being most of DEGS in L2PD (93%) shared in sPD. As compared to controls, we identified 437 DEGs in the PD group as a whole under an adjusted *P* below 0.05 (Fig 3A and B, and Table EV4). These findings are in line with two previous studies reporting expression changes associated with PD in DAn, at least with L2PD (sPD not studied) (Nguyen et al, 2011; Reinhardt et al, 2013). Upregulated DEGs (*n* = 254) were largely involved in neural functions and transcription factor regulatory activity, whereas downregulated DEGs (*n* = 183) affected basic homeostasis (Table EV5). One upregulated DEG was *SNCA* (> 2.5-fold), a gene involved in familial PD and sPD whose encoded protein, α-synuclein, aggregates in Lewy body inclusions which represent a hallmark of PD (Lang & Lozano, 1998a,b). Another upregulated DEG was *SYT11* (> 5-fold) which has been top-linked associated to PD across genomewide association studies (Nalls et al, 2014). We subsequently selected seven genes involved in neural functions from the top-30 list of upregulated DEGs (*OTX2*, *PAX6*, *ZIC1*, *SYT11*, *DCT*, *DCC*, and *NEFL*)

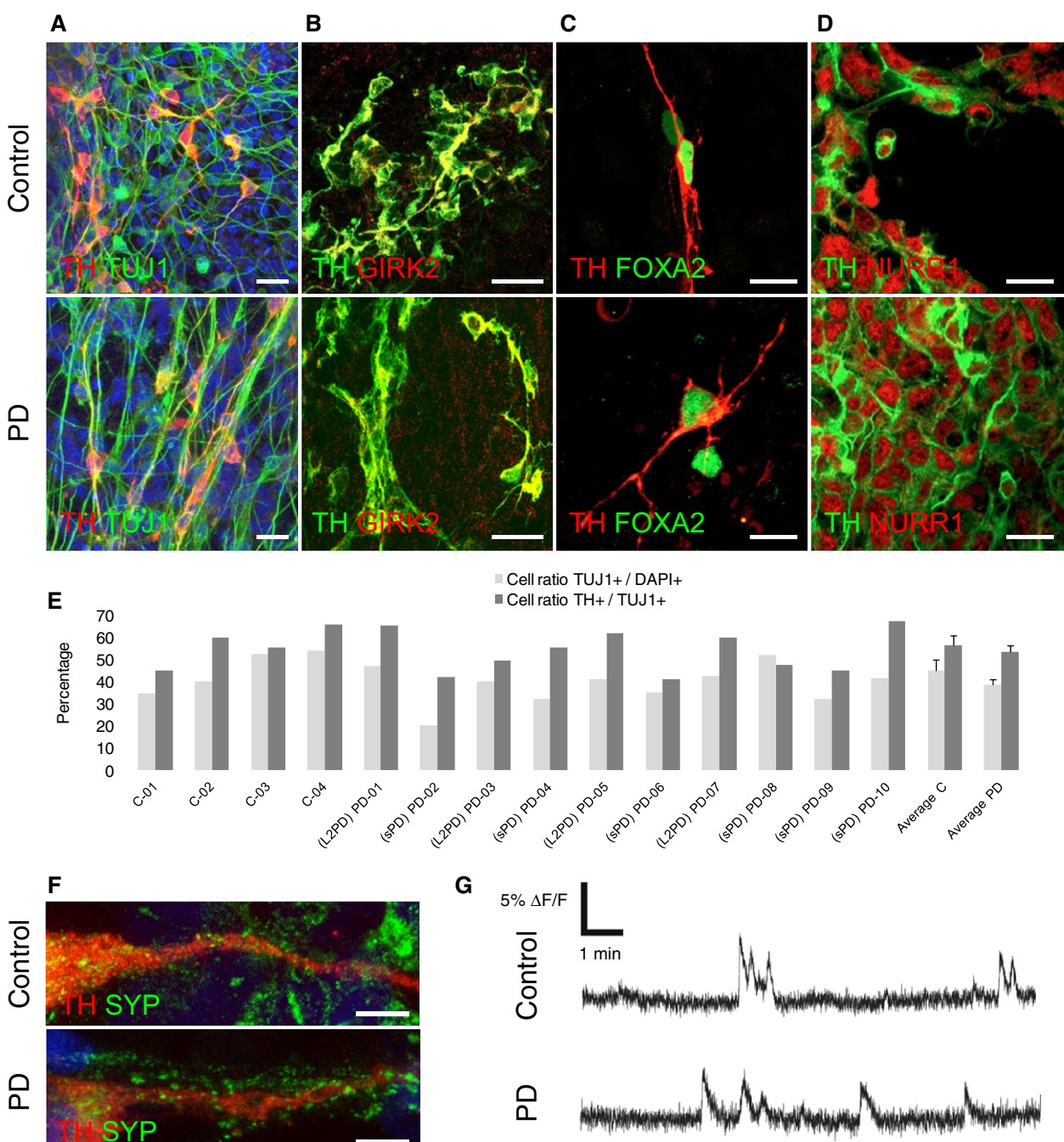

Figure 1.    Generation and characterization of iPSC-derived DAn.

A–G    After iPSC-based reprogramming of fibroblasts, we generated DAn from PD patients (*n* = 10) and gender- and age-matched healthy controls (*n* = 4) using a 30-day differentiation protocol. Blind to researcher, resulting iPSC-derived DAn showed similar properties and maturation state in PD and controls. Specifically, studied iPSC-derived DAn encompassed 30-day ventromedial (vm)-DAn-enriched cultures of morphologically mature DAn showing typical bipolar morphology, lacking PD neural phenotypes, mostly of the A9 subtype, and showing properties of tyrosine hydroxylase (TH)+ mature neurons. Representative immunocytochemical analyses of vmDAn from PD patients and controls showed that (A) TUJ1-positive cells expressing neural tubulin III (TUJ1) co-expressed TH (Scale bar, 25 μm); (B) co-expressed the A9 subtype marker of inward rectifier K+ channel GIRK2 (Scale bar, 25 μm); (C) co-expressed the A9 subtype marker of transcription factor Forkhead Box A2 (FOXA2) (Scale bar, 12.5 μm); and (F) co-expressed the synaptic vesicle protein synaptophysin (SYP) which was located along neurites (Scale bar, 5 μm). (D) The 1-week neural progenitor cells (NPCs) from which iPSC-derived DAn were generated strongly expressed DAn progenitor markers such as the nuclear-related receptor 1 NURR1 (Scale bar, 12.5 μm). (E) We observed similar and comparable cell counts of total neurons in PD and controls as quantified by TUJ-1/DAPI and of DAn as quantified by TH/TUJ1. Selected DAn cells were generated upon 2-6 iPSC lines per subject (See methods). Error bars indicate SEM. C, control; TUJ1, neuron-specific class III b-tubulin. (G) Calcium imaging assay revealed strong spontaneous neuronal activity. %ΔF/F indicates the relative change in fluorescence of the monitored neurons. Collectively, these results indicate that the iPSC-derived DAn used our study were sufficiently mature for the studied parameters to be functional and to spontaneously form active neural networks in a similar and comparable manner for PD and controls.

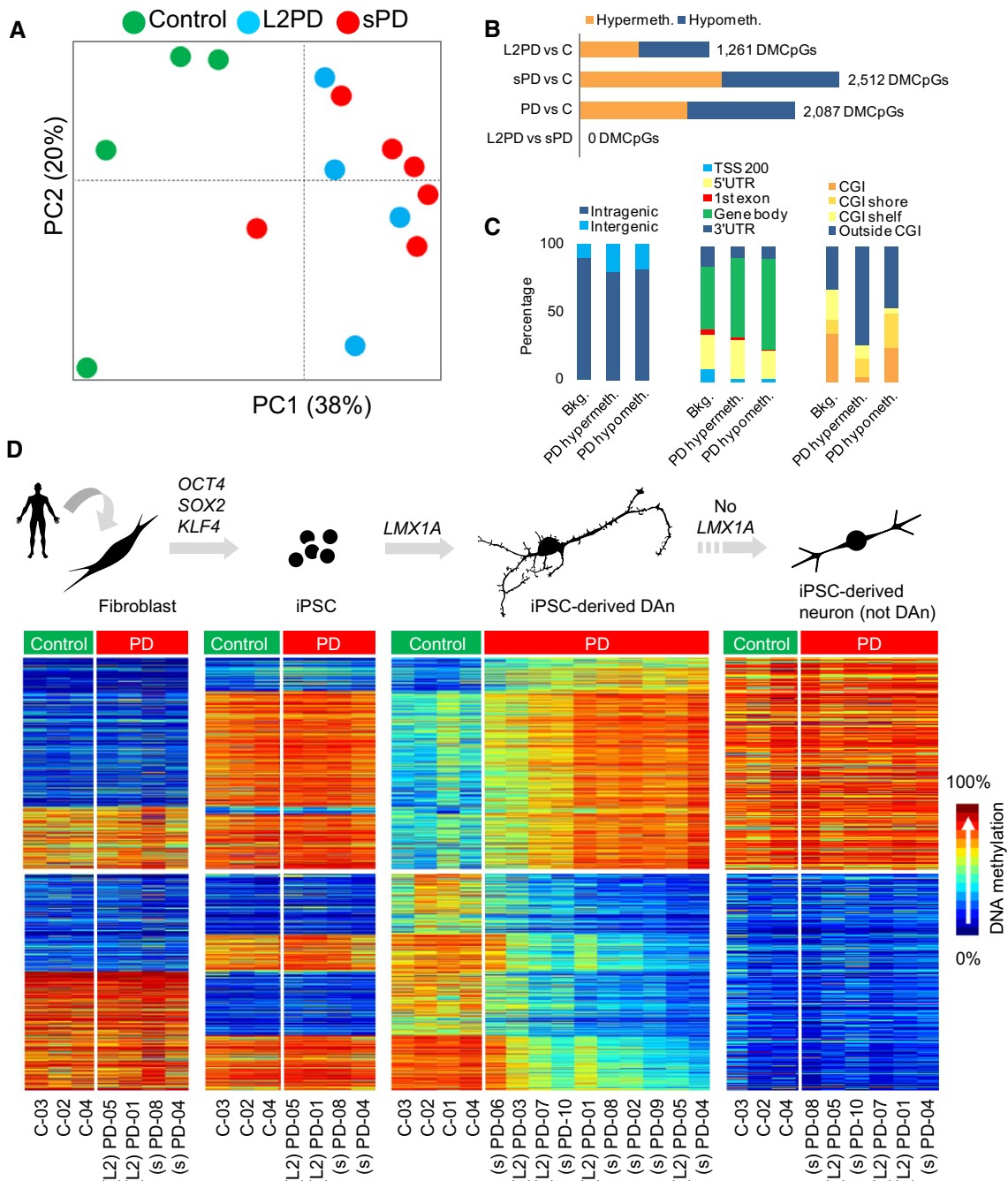

**Figure 2. DNA methylation analysis of iPSC-derived DAn from monogenic LRRK2-associated PD (L2PD) and sporadic PD (sPD) patients across the cell types involved in the cell reprogramming and differentiation processes.**

A Principal component analysis (PCA) of methylation data from 28,363 CpG sites with variable methylation values (SD > 0.1) in iPSC-derived DAn cell lines ($n$ = 14). This analysis shows different DNA methylation profiles between PD patients and controls, and no differences between L2PD and sPD. PC, principal component.

B Bar diagram showing the number of differentially methylated CpGs (DMCpGs) among iPSC-derived DAn lines ($n$ = 14) as detected in multiple comparisons. This analysis indicates that L2PD and sPD share similar DNA methylation changes with respect to controls. Hypermeth, hypermethylation; hypometh, hypomethylation.

C Relative distribution of DMCpGs in PD iPSC-derived DAn lines ($n$ = 10) across inter- and intragenic regions (left), across different gene-related regions (middle), and within CpG islands (CGI), CGI shores, CGI shelves, or outside CGI (right). Bkg, background (methylation platform).

D DNA methylation analysis across the cell types involved in the cell reprogramming and differentiation processes and also iPSC-derived neural cultures not-enriched-in-DAn. Scheme of the cell reprogramming and differentiation protocols (upper part of the panel). Heatmaps and density color code of the 2,087 DMCpGs identified in iPSC-derived DAn from PD patients with respect to controls ($n$ = 14), as well as their methylation status in parental fibroblasts ($n$ = 9), undifferentiated iPSCs ($n$ = 9), and iPSC-derived neurons (not-DAn) ($n$ = 9) (lower part of the panel) (Wilcoxon rank test for independent samples with delta-beta above |0.25| and FDR-adjusted $P$ < 0.05).

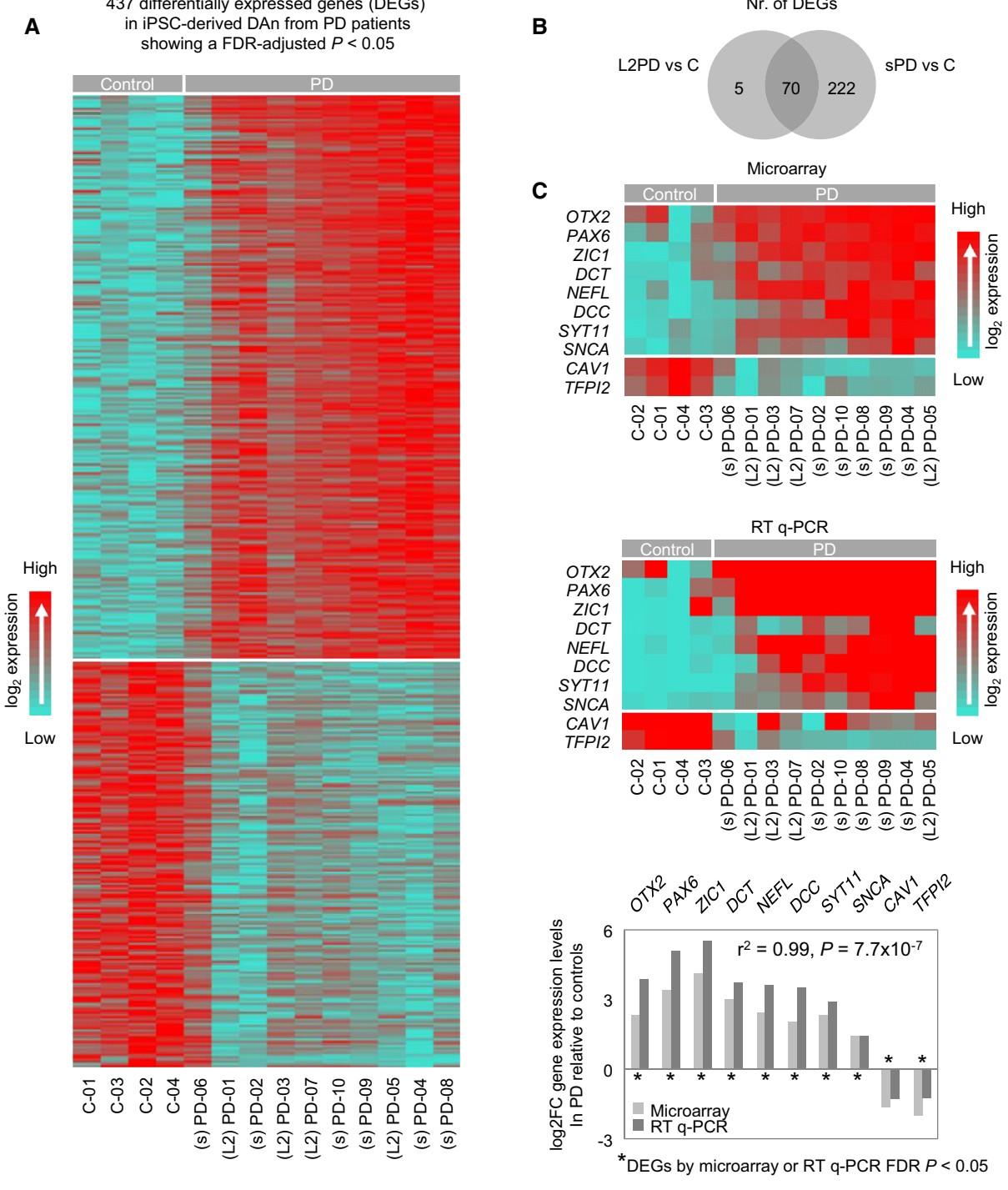

**Figure 3. Genomewide gene expression analysis of iPSC-derived DAn from PD patients and controls.**

A Heatmap showing the 437 differentially expressed genes (DEGs) detected in PD patients with either L2PD (L2) or sPD (s) as compared to controls (C) (n = 14), and density color code for gene expression levels. This analysis indicates that L2PD and sPD share similar gene expression changes with respect to controls. A total of 254 DEGs were upregulated in PD whereas 183 showed significant downregulation (Linear model with empirical bayes moderation of the variance similar to ANOVA with FDR-adjusted P < 0.05).

B Venn diagram representing the number of DEGs in iPSC-derived DAn from PD patients with respect to healthy subjects (n = 14) showing that most DMCpGs in L2PD were common to sPD.

C Technical validation of 10 top DEGs in PD iPSC-derived DAn as compared to controls (n = 14) identified by genomewide gene expression study (upper heat-map) and validated by real-time qPCR (lower heat-map). Relative gene expression and statistical significances were calculated using the ΔΔCt method and *GAPDH, ACTB,* and *PPIA* as endogenous controls. Pearson correlation analysis between microarray and real-time qPCR data showed a high degree of correlation ($r^2$ = 0.99, P = 7.7 × $10^{-7}$) (bottom graph). L2, *LRRK2*-associated PD; s, sporadic PD.

and *SNCA* to validate the array data by real-time qPCR (Fig 3C) and to study their protein expression levels by immunoblot. We detected a > 2-fold protein upregulation of all genes except *NEFL* (Fig EV3A). Moreover, protein expression of some DEGs co-localized at the single-cell level with the DAn marker tyrosine hydroxylase (Fig EV3C). These findings point toward the presence of gene and also protein expression changes in DAn from PD patients which occur simultaneously along with DNA methylation changes.

### PD DNA mehtylation changes are associated with gene expression

We then analyzed the relationship between gene expression and DNA methylation levels in iPSC-derived DAn from PD patients. We found a significant correlation in 17% of the 2,087 DMCpGs ($n = 353$) involving 239 different genes (Fig 4A and B). The percentage of correlation between methylation and expression was similar to that reported in other studies (Kulis *et al*, 2012). Of the 353 unique correlating DMCpGs, a total of 73% showed an inverse association of DNA methylation and expression, whereas the remaining 27% DMCpGs were positively associated (Table EV6). DMCpGs showing inverse association were situated in introns and 5′ untranslated regions, whereas those with positive association were enriched in introns (Fig 4C). Globally, DMCpGs showing association with gene expression were more frequently located in introns (49%) than in 5′ regions (24%). These findings suggest that, as reported in cell differentiation and cancer (Jones, 2012; Kulis *et al*, 2012, 2013), PD gene-body DNA methylation changes play a role in regulating gene expression, both inversely and positively.

### Differential DNA methylation in PD is enriched in enhancer elements

We further studied the potential molecular mechanisms underlying the association between DNA methylation and gene expression. We analyzed our data in the context of recently available functional chromatin states (Ernst *et al*, 2011). Hypermethylated DMCpGs were highly enriched in enhancer elements (35% vs. 12% in the background, $P = 0.3 \times 10^{-14}$), while hypomethylated DMCpGs were enriched in Polycomb-repressed regions (31% vs. 12% in the background, $P = 0.2 \times 10^{-14}$) (Fig 4D). When considering only the DMCpGs correlating with gene expression, DMCpGs with inverse correlation were located in promoters and enhancers, whereas DMCpGs with positive correlation were associated with repressed regions (Fig 4E). These results suggest that PD-associated DNA methylation changes target functionally active sequences.

### PD enhancer hypermethylation is associated with the downregulation of a TFs network

We next inquired whether the DMCpGs identified in PD patients were enriched for transcription factor (TF)-binding sites (TFBSs). To this end, we overlapped our data with TFBS clusters generated by ChIP-seq in the Encyclopedia of DNA Elements project (Dunham *et al*, 2012; Gerstein *et al*, 2012; Lee *et al*, 2012). We found enrichment for binding sites of 23 TFs in PD-hypermethylated DMCpGs and of only two TFs in PD-hypomethylated DMCpGs (Fig 5A and Table EV7). Since the PD-associated hypermethylation affected most

prominently to enhancer elements (35%) (Fig 4D), we focused on this chromatin state and observed that 65% of all PD hypermethylated enhancers became demethylated from iPSCs to DAn in controls, whereas only 8% significantly lost methylation in PD patients which overall retained higher methylation levels (Table EV3 and Fig 2D). Furthermore, among the 23 TFs showing TFBS enrichment, we found reduced gene expression of four TFs, namely *FOXA1*, *NR3C1*, *HNF4A,* and *FOSL2*, whose downregulation was also significantly associated with increased methylation levels at enhancer DMCpGs in PD (Fig 5B–D). Moreover, variable expression of these TFs together with the expression of other TFs was coordinated in PD DAn (Fig 6), suggesting that the expression level of a TF network, rather than that of individual TFs, was associated with the hypermethylation of enhancers in PD. In addition, *HNF4A* and *FOSL2* showed a significant downregulation of protein levels as detected by immunoblot, whereas *NR3C1* showed a downregulation trend which did not reach significance (Figs 7A and EV3B, and Source data for Fig 7). These data complement recent work linking TF binding to enhancers and tissue-specific hypomethylation (Stadler *et al*, 2011; Hon *et al*, 2013; Xie *et al*, 2013; Ziller *et al*, 2013), and also downregulation of TF networks to enhancer hypermethylation (Agirre *et al*, 2015). Our findings suggest that the incomplete epigenetic remodeling observed for the 2,087 DMCpGs identified in iPSC-derived DAn from PD patients might be mediated by the aberrant downregulation of a network of key TFs whose deficiency could prevent their target sites to become demethylated during the differentiation from iPSCs to DAn.

## Discussion

We report the first genomewide DNA methylation study of iPSC-derived DAn in PD. We found that DAn from PD patients are epigenetically altered with respect to healthy controls and that DAn from monogenic L2PD and sPD share similar DNA methylation abnormalities. These methylation changes cannot be explained by technical biases since all iPSC-derived DAn lines were equally generated and characterized in parallel, blind to the researcher, and differentiated DAn had similar morphological and functional properties (Sanchez-Danes *et al*, 2012b). We also found that the PD-associated methylation changes were not present in parental skin cells or undifferentiated iPSCs and become uncovered only upon differentiation into the DAn cells targeted in PD. In addition, the methylation profile of DAn from PD patients, with either monogenic or sporadic PD, resembled that of neural cultures not-enriched-in-DAn indicating a failure to fully acquire the epigenetic identity own to healthy DAn in PD. We also detected that gene and protein expression changes occur simultaneously along with methylation alterations in PD and that these alterations are similar in monogenic L2PD and sPD. Finally, we found that the DNA methylation changes present in PD DAn are partially associated with gene expression, target gene regulatory sequences such as enhancers, and correlate with the RNA and protein downregulation of a network of TFs.

The lack of DNA methylation differences between DAn from patients with L2PD and sPD provides the proof-of-concept of common or at least converging epigenomic changes in these disease forms (Zhu *et al*, 2011). However, one limitation of our work is that we studied only one monogenic form of PD (L2PD) and therefore

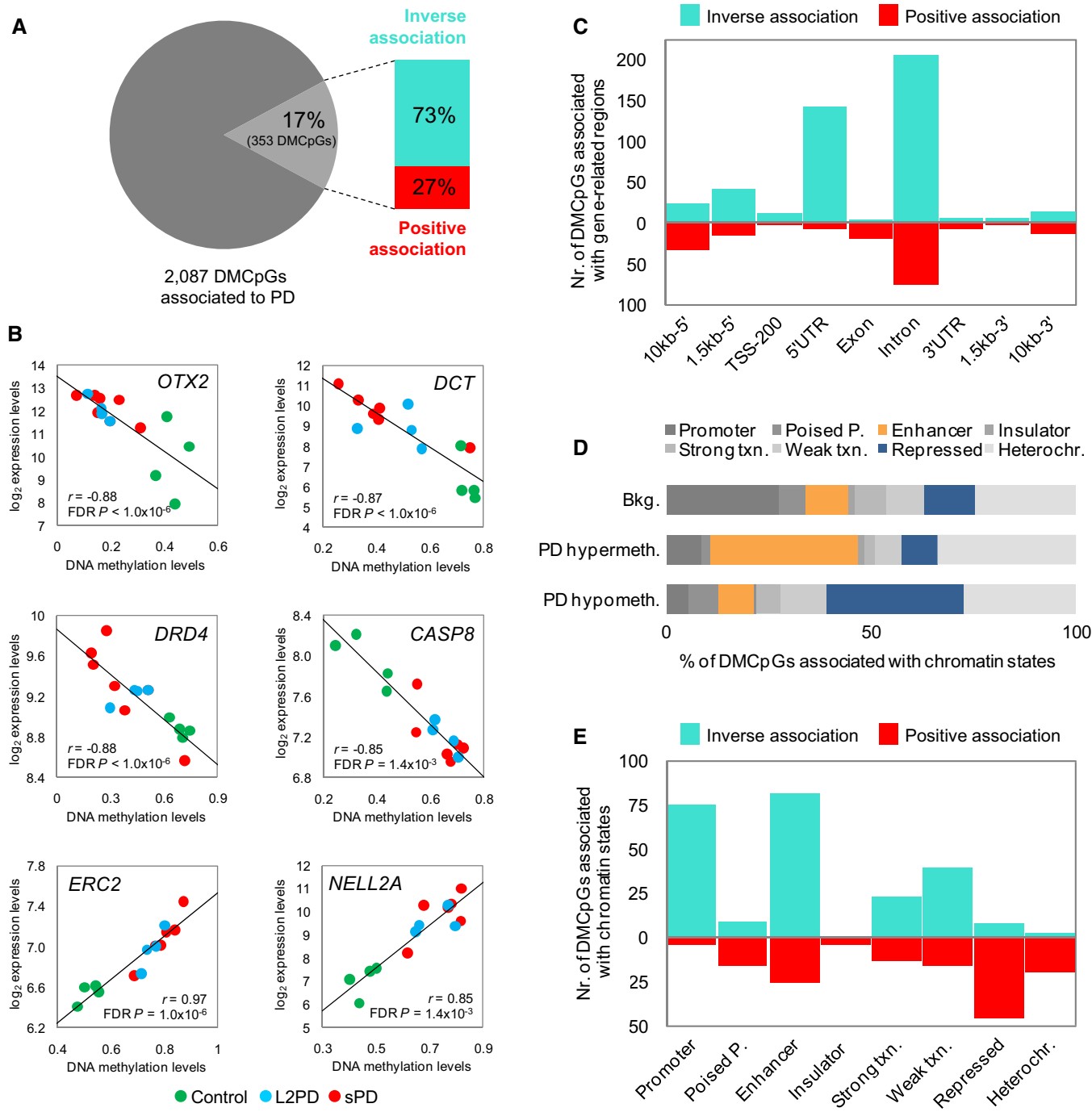

**Figure 4.  Correlation between DNA methylation values of differentially methylated CpGs (DMCpGs) and expression levels of the associated genes in PD iPSC-derived DAn.**

A  Graphical representation of the percentage of DMCpGs associated with gene expression.

B  Examples of genes showing significant correlation including a brief functional description: *OTX2*, transcription factor for midbrain DAn development which blocks *LMX1A*; *DCT*, dopachrome tautomerase, induced by *OTX2*, is a detoxifying enzyme of DA metabolites involved in the synthesis of neuromelanin; *DRD4*, DA receptor type 4; *CASP8*, apoptosis extrinsic pathway member inducing *CASP3*; *ERC2*, cytoskeleton organizer at nerve terminals for neurotransmission; *NELL2*, neural epidermal growth factor-like 2 involved in neural differentiation.

C  Counts of DMCpGs associated with gene expression across gene-related regions.

D  Relative distribution of DMCpGs across chromatin states. Poised P, poised promoter; Txn, transcription; Heterochr, heterochromatin.

E  Counts of DMCpGs associated with gene expression across chromatin states.

Data information: In (A) and (B), *n* = 14 cell lines, Spearman correlation analysis with FDR-adjusted *P* < 0.05.

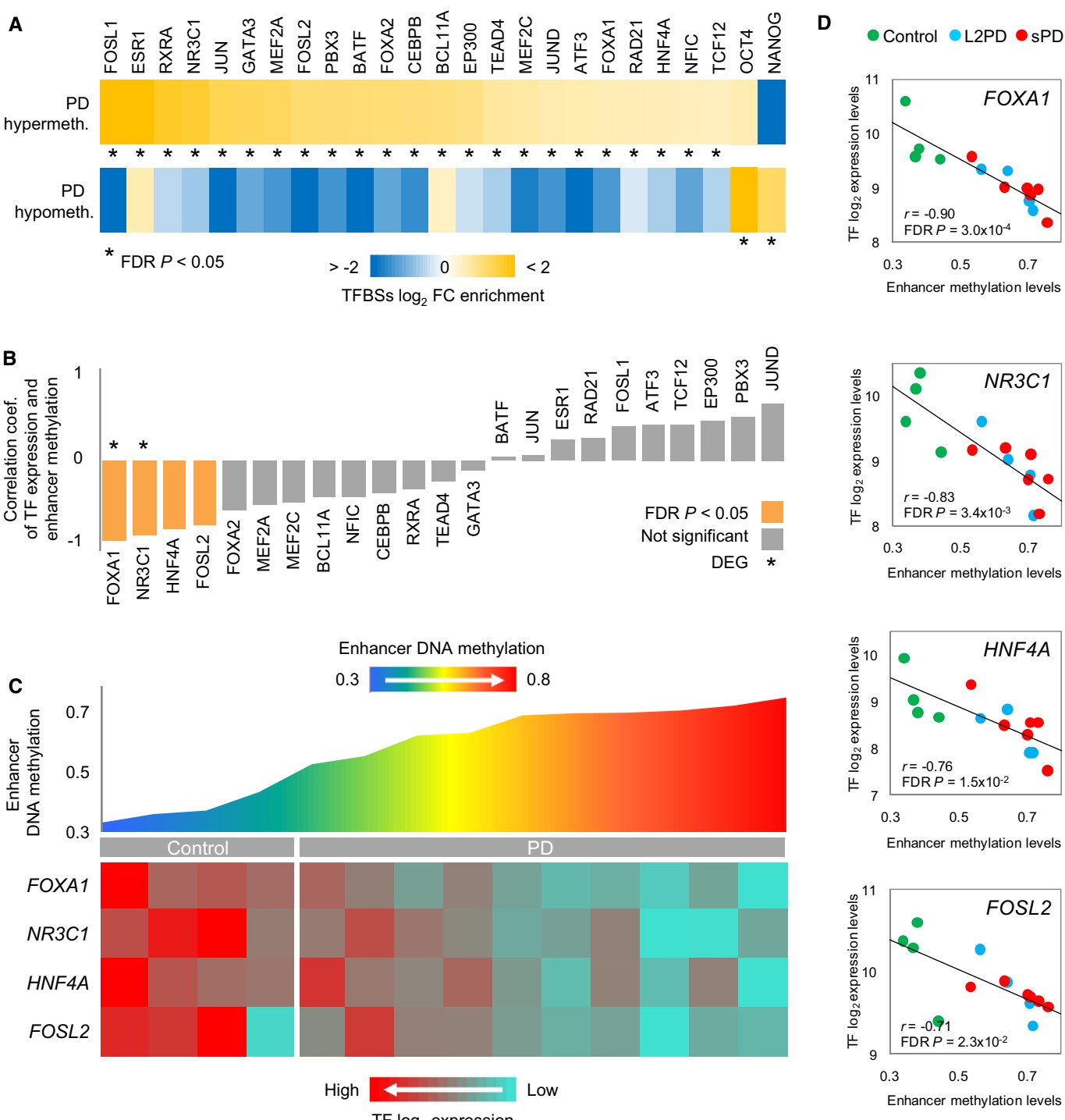

**Figure 5.  Association between gene expression levels of transcription factors (TFs) and DNA methylation levels at PD-hypermethylated enhancers from PD iPSC-derived DAn.**

A   Relative enrichment of TF-binding sites (TFBSs) overlapping with the 2,087 DMCpGs detected in PD iPSC-derived DAn ($n$ = 10). This analysis shows an enrichment of binding sites for 23 TFs in regions hypermethylated in PD.

B   Bar plot showing the results of the Spearman correlation analysis between levels of TF gene expression and of average DNA methylation at the 376 enhancer sites hypermethylated in PD iPSC-derived DAn ($n$ = 10). Coef, coefficient.

C   Graphical representation of average methylation at PD-associated enhancers and gene expression of the key TFs *FOXA1*, *NR3C1*, *HNF4A*, and *FOSL2* in iPSC-derived DAn ($n$ = 14).

D   Scatter plots of key TF gene expression and DNA methylation levels at enhancers in iPSC-derived DAn ($n$ = 14).

Data information: Fisher's exact test with a FDR-adjusted $P$ < 0.05 in (A), and Spearman correlation analysis with FDR $P$ < 0.05 in (B) and (D).

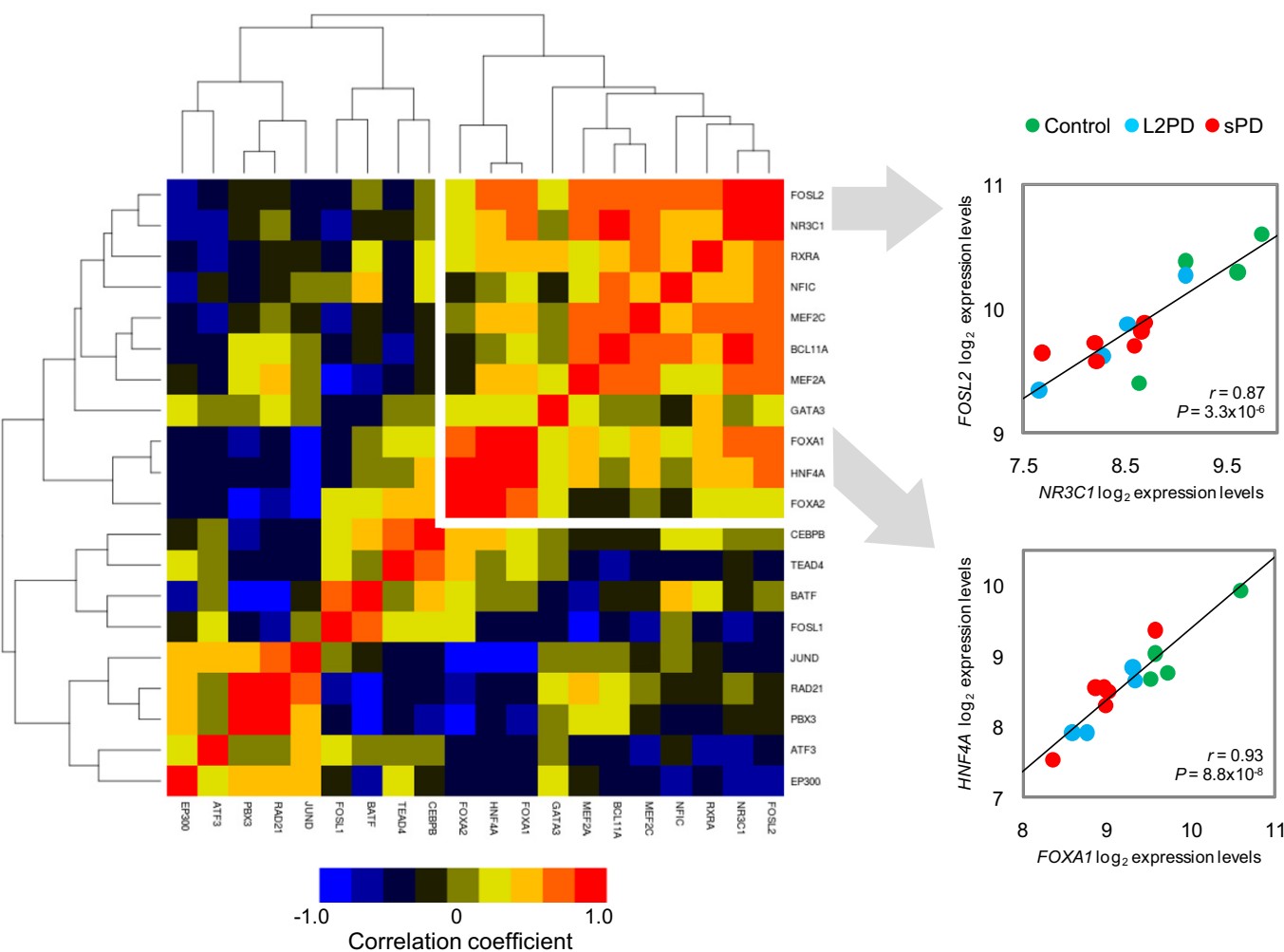

**Figure 6.  Correlation matrix of gene expression levels among TFs showing TF-binding sites (TFBS) enrichment at PD-hypermethylated enhancers.**
Pearson correlation coefficients and correlation *P*-values were calculated evaluating TF gene expression levels in iPSC-derived DAn from PD patients and controls (*n* = 14) under a 2-tailed Student's *t*-test. Among TFs showing coordinated expression, maximal correlation was observed for *FOXA1*, *FOXA2*, *NR3C1*, *HNF4A*, and *FOSL2*.

the epigenetic involvement in other PD familial forms remains to be explored. Our findings are compliant with the observation that L2PD is clinical and neuropathologically similar to sPD lacking *LRRK2* mutations (Healy *et al*, 2008). Yet the interpretation of our results requires further considerations. First, the epigenome is expected to reflect the influence from the environment and thus one possibility is that unknown environmental factors could trigger epigenomic changes in PD patients. This environmental contribution would be expectable for sPD (Feil & Fraga, 2011), but the age-dependent reduced penetrance of the *LRRK2* G2019S mutation together with its identification not only in monogenic but also in sPD cases (Healy *et al*, 2008) could also support an environmental involvement in L2PD (Farrer, 2006; Urdinguio *et al*, 2009). Second, an alternative possibility is that the genetic background from PD patients could drive DNA methylation changes not only in monogenic L2PD as expectable but also in sPD. In this regard, it has been described that sPD could be caused by an accumulation of common polygenic alleles with relatively low effect sizes (Escott-Price *et al*, 2015). Thus, a possible integrative interpretation of our findings is

that cumulative genetic risk factors such as single nucleotide polymorphisms (SNPs) or copy number variants (CNVs) in sPD, or *LRRK2* mutations in L2PD, alone or in combination with still unknown environmental factors could be underlying the common epigenetic changes detected in both disease forms. These environmental and/or genetic factors arising from different but converging pathways could ultimately cause common end-point alterations in L2PD and sPD.

The PD epigenetic changes were present only in iPSC-derived DAn but not in undifferentiated iPSC, in fibroblasts, nor in iPSC-derived neural cultures not-enriched-in-DAn. These results indicate that the PD-associated DNA methylation changes are specific of DAn but not of other types since neural cultures not-enriched-in-DAn did not reveal methylation differences. They also suggest that molecular defects, of an environmental and/or genetic origin as discussed above, should be carried from the PD patient, latent in fibroblasts, and manifested only upon differentiation into DAn. Here it should be mentioned that although in PD when and where the degenerative process starts is unclear, and the related underlying

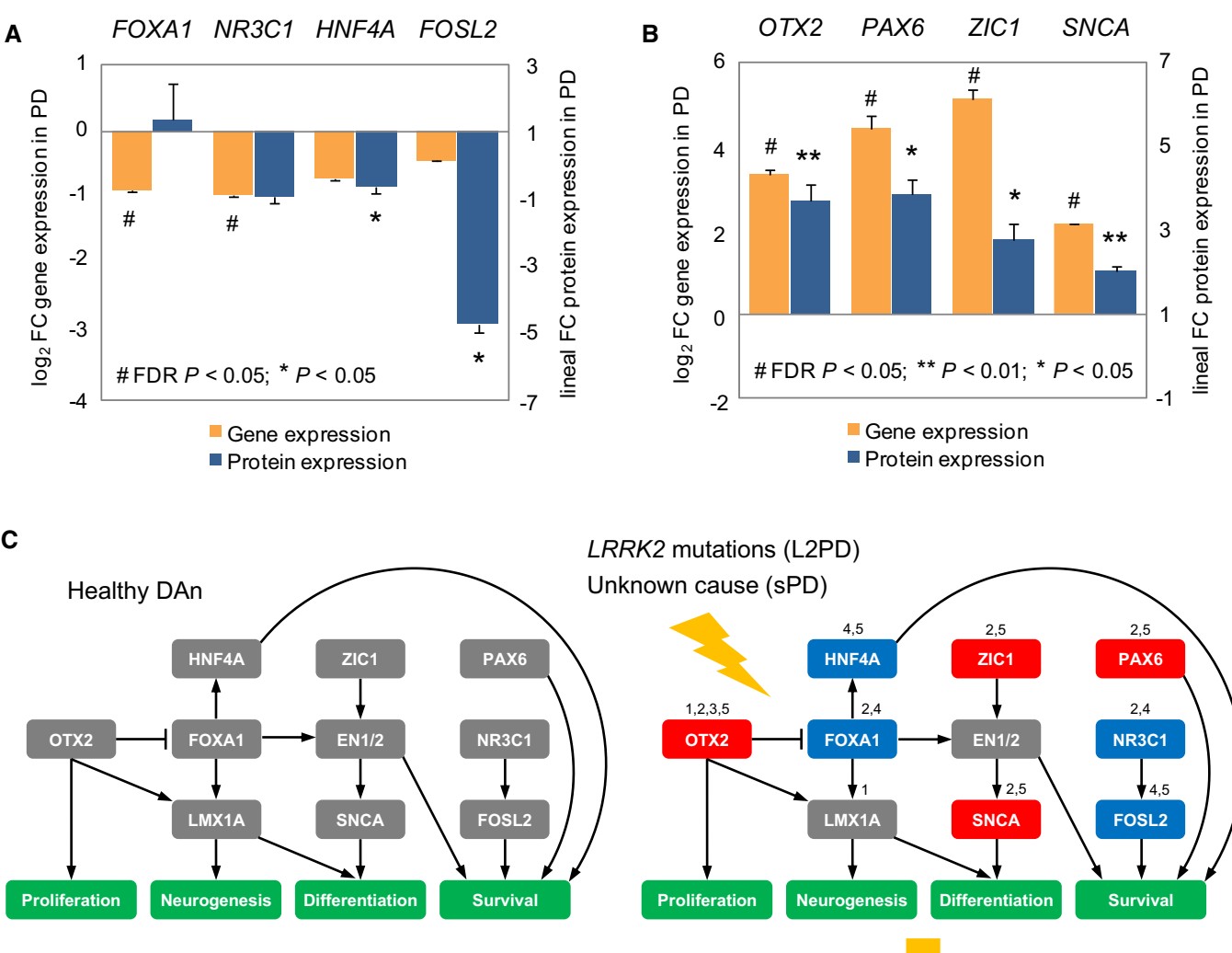

**Figure 7. Gene and protein expression of PD downregulated key TFs and proposed theoretic model.**

A, B   Graphical representation of gene and protein expression of the key TFs *FOXA1, NR3C1, HNF4A,* and *FOSL2* (A) (see also Fig EV3B and Source data for Fig 7), or of the PD upregulated genes (B) (see also Fig EV3A and Source data for Fig 7) identified in iPSC-derived DAn from PD patients, normalized to the expression of controls, and expressed, respectively, as log₂ or lineal fold change (FC) values. (For gene expression, linear model with empirical bayes moderation of the variance similar to ANOVA with FDR-adjusted *P* < 0.05; for protein expression, two-tailed Student's *t*-test (**P* < 0.01, **P* < 0.05). Samples were studied at least in three independent experiments. Data are represented as group mean ± SEM).

C      Proposed model in which deficits of the key TFs *FOXA1, NR3C1, HNF4A,* and *FOSL2* relevant to PD lead to DNA methylation changes. Key 1 denotes differentially methylated gene, key 2 denotes DEG, key 3 denotes significant correlation of DNA methylation with proximal gene expression, key 4 denotes significant correlation of key TF gene expression with distal DNA methylation at enhancers, and key 5 denotes differentially expressed protein. Blue-shaded boxes indicate gene expression downregulation, whereas red-shaded boxes indicate upregulation. Illustration of our model was built by selecting most prominent alterations detected by unbiased genomewide approaches in our study.

Source data are available online for this figure.

biological changes remain unknown, several works have reported molecular defects in fibroblasts from L2PD and also from other PD monogenic forms (Hoepken *et al*, 2008; Rakovic *et al*, 2013; Ambrosi *et al*, 2014; Yakhine-Diop *et al*, 2014). In this context, our findings are compatible with the emerging notion of PD as a systemic disease affecting other tissues apart from the nervous system (Beach *et al*, 2010; Shannon *et al*, 2012).

The methylation profile from PD DAn resembled that of neural cultures not-enriched-in-DAn indicating a failure in PD to fully acquire the epigenetic identity own to healthy DAn. The methylation changes identified in PD occurred in spite of the apparently normal dopaminergic phenotypes observed in our early 30-days DAn cultures which consisted in functionally and morphologically mature DAn which were similar in PD and controls

(Sanchez-Danes *et al*, 2012b). This is in agreement with previous studies describing normal dopaminergic phenotypes in PD iPSC-derived DAn (Byers *et al*, 2011; Nguyen *et al*, 2011; Cooper *et al*, 2012; Rakovic *et al*, 2013; Reinhardt *et al*, 2013) with the exception of one Parkin study reporting altered dopamine release and uptake (Jiang *et al*, 2012). However, we found that the epigenetic changes detected in our early 30-days DAn cultures antedated late (75-days), spontaneous (not-drug induced), PD-associated phenotypes which were previously described in our model and included impaired axonal outgrowth, deficient autophagic vacuole clearance, and accumulation of α-synuclein (Sanchez-Danes *et al*, 2012b; Orenstein *et al*, 2013). Consistent with our methylation findings, these PD phenotypes were also common in DAn from L2PD and sPD patients (Sanchez-Danes *et al*, 2012b). Yet although the early DNA methylation changes reported here antedated these long-term PD phenotypes potential causality needs to be explored in future works.

As reported in cell differentiation and cancer (Jones, 2012; Kulis *et al*, 2012, 2013), we found that the identified epigenetic changes correlate partially with expression, indicating that PD gene-body DNA methylation changes play a role in the regulation of gene expression. We also found that the PD DNA hypermethylation changes are prominent at enhancer regulatory regions. In line with other studies (Agirre *et al*, 2015), we also found that in PD, the hypermethylation of enhancers appears to be related to the downregulation of a network of key TFs, rather than of individual TFs. These data complement recent work linking TF binding to enhancers and tissue-specific hypomethylation (Stadler *et al*, 2011; Hon *et al*, 2013; Xie *et al*, 2013; Ziller *et al*, 2013) and suggest a theoretic model in which the incomplete epigenetic remodeling in PD DAn might be related to the downregulation of a network of key TFs whose deficiency could prevent their binding sites to become demethylated during the differentiation from iPSCs to DAn. Key TFs from this network have been previously associated with the specification of the substantia nigra (*FOXA1*, *NR3C1*, and *HNF4*) and of fetal brain (*FOSL2*) (Ziller *et al*, 2013). Of these, *FOXA1* is involved in maintaining the dopaminergic phenotype in adult mesodienchepalic DAn (Stott *et al*, 2013; Domanskyi *et al*, 2014), whereas the glucocorticoid receptor *NR3C1* regulates DAn neurodegeneration in PD (Ros-Bernal *et al*, 2011). Downregulation of *HNF4A* has been reported in blood from PD patients as a PD biomarker correlating with motor symptoms severity (Potashkin *et al*, 2012; Santiago & Potashkin, 2015), whereas *FOSL2* has been linked to dyskinesia, a major side effect in the DA substitutive treatment with L-DOPA (Cao *et al*, 2010). Since these TFs seem to be relevant to the development and maintenance of midbrain DAn in PD (Lahti *et al*, 2011, 2012), their deficiency might be related to impairments in the maintenance of a differentiated DAn epigenetic cellular identity in PD (Holmberg & Perlmann, 2012). In this theoretic model, the upregulation of gene and protein expression of other TFs (*OTX2*, *PAX6*, or *ZIC1*) and genes (*SNCA*, *DCC*, or *DCT*) (Figs 7B and EV3A, and Source data for Fig 7) might transiently circumvent deficits of the network of key TFs, at least in early culture stages.

One hypothesis proposes the neurodevelopmental origin of neurodegenerative diseases in the sense that molecular mechanisms occurring during development are abnormally recapitulated in neurodegenerative disease processes (Goedert *et al*, 1993; Schafer & Stevens, 2010). For example, a recent study showed that neural degeneration in Alzheimer's disease involves the abnormal re-activation of cellular self-destruction mechanisms which take place during neural development (Nikolaev *et al*, 2009). Since we found that the epigenetic pattern from PD iPSC-derived DAn showed certain similarities to neural cultures not-enriched-in-DAn, our data could be interpreted in light of this theory pointing towards the presence of possible developmental epigenetic defects associated with PD. In addition, developmental deficits of TFs which are key in the differentiation of DAn have been recently linked to PD (Laguna *et al*, 2015).

We showed that iPSC-derived DAn from PD patients exhibit epigenomic and transcriptomic alterations, and therefore, our findings may have implications for future cell replacement therapies. The use of pluripotent stem cells has proved as a valuable *in vitro* system to investigate disease, as well as a promising tool for brain transplantation and dopaminergic restoration in PD. Yet the generation of DAn for cell replacement is still matter of research (Lindvall & Bjorklund, 2011). Recently, the autologous transplantation of iPSC-derived neurons into the striatum of healthy monkeys has been suggested to be advantageous as compared to allogenic grafts, at least in terms of reduced immunogenicity (Morizane *et al*, 2013). Our study suggests that future cell therapeutic strategies should pay attention to the correct epigenomic status of the reprogrammed dopaminergic cells, especially when using patient own cells. In summary, using a patient-specific iPSC-based DAn system, our study provides the first evidence that epigenetic deregulation is associated with both monogenic and sporadic PD.

# Materials and Methods

### PD patients and generation of iPSC-derived DAn

We used mature iPSC-derived DAn lines of 30-days of differentiation generated and characterized in parallel blind to researcher previously (Sanchez-Danes *et al*, 2012b) using a published protocol (Sanchez-Danes *et al*, 2012a). Expanded subject information, cell characterization, and technical details are extensively described in these precedent studies where they should be consulted. Of these, a summary presented in Table 1, Fig 1, and here as it follows. Arm surface skin biopsies of 3 mm of diameter were obtained from *LRRK2* PD patients carrying the G2019S mutation (L2PD, $n = 4$); sporadic PD patients lacking family history of PD and mutations in known PD genes (sPD, $n = 6$); and gender- and age-matched healthy individuals without neurological disease history (controls, $n = 4$). Primary cultures of keratinocytes or fibroblasts were reprogrammed using retroviral delivery of *OCT4*, *KLF4*, and *SOX2* to generate 2–6 iPSC lines per individual ($n = 50$ lines). Of these, we selected the two more homogenous lines per subject which were thoroughly characterized and shown to be fully reprogrammed to pluripotency. For the directed differentiation of iPSC to ventral midbrain dopaminergic neurons (vmDAn), we used a 30-days protocol based on the lentiviral-mediated forced expression of the vmDAn determinant *LMX1A* together with DAn patterning factors and co-culture with mouse PA6 feeding cells to provide trophic factor support. DAn

pellets were obtained by mechanic separation of the PA6 layer with a finely drawn Pasteur pipette. Differentiated cells were 30-days vmDAn-enriched cultures of morphologically fully mature DAn lacking PD neural phenotypes (Sanchez-Danes *et al*, 2012b), mostly of the A9 subtype which showed properties of TH[+] mature neurons: (i) electrophysiological analysis showed action potentials, (ii) the majority of DAn presented the typical bipolar morphology, (iii) co-expressed the A9 subtype marker GIRK2, and (iv) co-expressed the DA transporter (DAT). Collectively, studied iPSC-derived DAn were sufficiently mature, in a similar and comparable manner for both PD and control cultures, to be functional and to spontaneously form active neural networks.

### Calcium fluorescence imaging of iPSC-derived DAn

We used the cell-permeant fluorescence dye Fluo-4-AM (Life Technologies) in combination with an imaging device to monitor spontaneous activity in PD and control cells. This fluorescence probe becomes active upon binding with calcium, therefore signaling the occurrence of action potentials in the neurons. Cultures of either PD or control cells were cultured in 30-mm-diameter petri dishes for 30-days and their activity monitored as follows. Prior to imaging, the culture dish was first washed with 4 ml PBS at room temperature to remove the original culture medium. The culture was next incubated for 30 min in a solution that contained 1 ml of recording medium (RM, consisting of 128 mM NaCl, 1 mM CaCl$_2$, 1 mM MgCl$_2$, 45 mM sucrose, 10 mM glucose, and 0.01 M Hepes; treated to pH 7.4) and 4 μg/ml of Fluo-4. The culture was then washed with fresh RM to remove residual free Fluo-4, and finally, a volume of 4 ml of RM was left in the dish for actual recordings. The culture dish was mounted on a Zeiss inverted microscope equipped with a CMOS camera (Hamamatsu Orca Flash 2.8) and an arc lamp for fluorescence. Gray-scale images of neuronal activity were acquired at intervals of 50 ms, and in a field of view that contained on the order of 100 cells. Cultures were visualized for 30 min at room temperature. The acquired data was next analyzed to extract the fluorescence traces for each neuron. In order to compare the fluorescence traces among neurons and cultures, the raw fluorescence signal $F(t)$ of a given neuron was normalized as $F^*(t) = \%\Delta F/F = 100 * (F(t) - F0)/F0$, where $F0$ is the fluorescence signal of the neuron at rest. Spontaneous neuronal activations were revealed in the fluorescence signal as a sharp increase of typically a 10% respect to the resting state.

### Generation of iPSC-derived neural cultures not-enriched-in-DAn

iPSC-derived neural cultures not-enriched-in-DAn were generated as technical controls from a subset of PD patients ($n = 6$) and healthy subjects ($n = 3$) using the previously described protocol (Sanchez-Danes *et al*, 2012a) but omitting the lentiviral expression of the vmDAn determinant *LMX1A* and DAn patterning factors. The lentiviral delivery of LMX1A results in a > 4-fold enrichment of the total final number of DAn to ~30% as compared to only ~7% without LMX1A (Sanchez-Danes *et al*, 2012a). Embryoid body (EB) formation was induced for forced aggregation and maintained in suspension in the presence of MEF condition medium for 3 days. EBs were then maintained for 10 days in ultralow attachment plates in N2B27 medium in the presence of FGF2. After that,

EBs were transferred to matrigel-coated plastic chamber slides and cultured in the differentiation medium in the absence of FGF2 for 30-days. The medium for each condition was changed every other day.

### DNA, RNA, and protein isolation

DNA, RNA, and proteins were isolated from one million cells using the Allprep DNA-RNA-Protein kit (QIAGEN). Concentration and quality of DNA and RNA were, respectively, determined in a Nanodrop 1000 Spectrophotometer and in an Agilent 2100 Bioanalyzer. Total protein concentration was assessed with the BCA assay (Thermo Scientific).

### DNA methylation analysis

The EZ DNA Methylation Kit (Zymo Research) was used for bisulfite conversion of 1,000 ng genomic DNA. Bisulfite-converted DNA was hybridized onto the Infinium Human Methylation 450K BeadChip Kit (Illumina) which covers 99% of the RefSeq genes and 96% of CpG islands at a single-base resolution (refer to Product Datasheet: http://www.illumina.com/products/methylation_450_beadchip_kits.ilmn). We have previously demonstrated a correlation coefficient of findings above 95% between the Illumina 450k platform and whole-genome bisulfite sequencing (Kulis *et al*, 2012). The Infinium methylation assay was carried out following criteria requested for this platform (Bibikova *et al*, 2011). All the array experiments were performed simultaneously at the same day, and PD and control samples were randomized at each BeadChip, ruling out any potential batch effect. Array data were analyzed by minfi package available through Bioconductor (Kasper Daniel Hansen and Martin Aryee. Minfi: Analyze Illumina's 450k methylation arrays. R package version 3.0.1). To exclude technical biases, we used an optimized pipeline with several filters developed at the Unidad de Hematopatología at IDIBAPS (Kulis *et al*, 2012). From the initial dataset of 485,512 sites (excluding probes detecting SNPs), we removed those with poor detection *P*-values ($P > 0.01$, $n = 670$) and those with sex-specific DNA methylation ($n = 6,614$). The remaining 478,228 sites were used for downstream analyses. Single CpG quantitative methylation values (beta values) in each sample were calculated as the ratio of the methylated signal intensity to the sum of methylated and unmethylated signals. Resulting quantitative methylation values ranged from 0, fully unmethylated, to 1, fully methylated CpGs. DNA from pure mouse PA6 cell cultures was included as a control sample that did not hybridize at readable levels in the human Illumina 450k methylation array, therefore not influencing methylation findings.

### Genomic annotation of CpGs

The 450k Human Methylation Array data was annotated using the current hg19 version of the UCSC Genome Browser as reference sequence. We assigned the CpGs into discrete categories according to their location with respect to gene-related regions: 10-kb 5′ region (10,000 to 1,501 bp upstream of the transcriptional start site (TSS)), 1.5-kb 5′ region (1,500 to 201 bp upstream of the TSS), TSS 200 (200 to 1 bp upstream of the TSS), 5′ UTR, gene first exon, gene

body (from the first intron to the last exon), 3′ UTR, 1.5-kb 3′ region (1 to 1,500 bp downstream of the 3′UTR), 10-kb 3′ region (1,501 to 10,000 bp downstream of the 3′UTR), and intergenic regions. Owing to the presence of alternative transcription start sites and regions containing more than one gene, some CpGs were assigned multiple gene annotations. For the DMCpGs location relative to a CpG island (CGI), we used the categories: within CGI, in CGI shore (up to 2 kb from the CGI edge), in CGI shelf (from 2 to 4 kb from the CGI edge), and outside CGI.

**Statistics: identification of differentially methylated CpGs**

For each CpG, we computed the difference between the DNA methylation level in the two groups under comparison. Subsequently, CpGs were considered differentially methylated (DMCpGs) when showing a delta-beta methylation difference above $|0.25|$ and a false discovery rate (FDR)-adjusted $P$ Wilcoxon rank test for independent samples below 0.05. A delta-beta above $|0.20|$ is detectable with a 99% confidence in our methylation platform (Bibikova *et al*, 2011). Consistently, the same cutoff of delta-beta above $|0.25|$ and FDR-adjusted $P < 0.05$ was used in our study to identify statistically significant methylation differences (DMCpGs) between cases and controls in iPSC-derived DAn, fibroblasts, undifferentiated iPSC, and iPSC-derived neural cultures not-enriched-in-DAn for any given comparison.

**Statistics: Gene expression analysis and identification of differentially expressed genes**

We hybridized 100 ng of RNA onto the Genechip Human Exon 1.0 ST Array (Affymetrix) which covers > 96% of the human transcriptome. Using a robust multi-array analysis (RMA) algorithm (Irizarry *et al*, 2003), pre-processing of microarray data resulted in single gene $\log_2$-transformed values from 36,079 transcripts (22,014 Entrez RefSeq. genes). We filtered out (i) genes with group mean signals < 50 percentile of all signals, and (ii) genes with standard deviation (SD) < 50 percentile of all SDs. The remaining 4,686 Entrez RefSeq genes were used for downstream analysis. RNA from pure mouse PA6 cell cultures was included as a control sample that did not hybridize at readable levels in the Affymetrix human array, therefore not influencing expression findings. We used the Bioconductor Limma package to detect differentially expressed genes (DEGs) under a linear model with empirical bayes moderation of the variance similar to ANOVA developed to adjust for small sample size in microarray studies (Smyth, 2004). A FDR-adjusted $P < 0.05$ was used to identify DEGs.

**Real-time quantitative RT–PCR**

We analyzed by real-time qPCR the expression levels of ten DEGs identified by the genomewide gene expression microarray: *OTX2, PAX6, ZIC1, DCT, NEFL, DCC, SYT11, SNCA, CAV1,* and *TFPI2* (Fig 3). cDNA was synthesized using the High Capacity cDNA Reverse Transcription kit (Applied Biosystems). We used 1 ng of cDNA for each real-time qPCR. Gene amplification was done using pre-designed Taqman Gene Expression assays in a StepOnePlus Real-time PCR System (Applied Biosystems). The $\log_2$ fold change (FC) values and statistical significances of DEGs were calculated using the ΔΔCt

method in the DataAssist v3.0 software (Applied Biosystems). We normalized the expression of DEGs to the housekeeping genes glyceraldehydes-3-phosphate dehydrogenase (*GAPDH*), beta-actin (*ACTB*), and cyclophylin A (*PPIA*). Pearson correlation between microarray and real-time qPCR data was computed using the SPSS 16.0 software. Commercially available assay numbers are as it follows: *OTX2* (hs00222238_m1), *PAX6* (hs00240871_m1), *ZIC1* (hs00602749_m1), *DCT* (hs01098278_m1), *NEFL* (hs00196245_m1), *DCC* (hs0018043 7_m1), *SYT11* (hs00383056_m1), *SNCA* (hs01103383_m1), *CAV1* (hs00971716_m1), *TFPI2* (hs00197918_m1), *GAPDH* (hs0275899 1_g1), *ACTB* (hs01060665_g1), and *PPIA* (hs04194521_s1).

**Biological enrichment analysis**

To determine whether genes associated with DMCpGs (1,178 genes/ 2,087 CpGs) or DEGs ($n = 437$) in PD were enriched in particular gene ontology (GO) terms, we used the Functional Annotation Tool (FAT) of the Database for Annotation, Visualization and Integrated Discovery (DAVID) (Huang da *et al*, 2009). Multiple hypothesis test correction was performed using the Benjamini & Hochberg algorithm (Benjamini, 1995).

**Correlation of DNA methylation and expression data**

Based on common gene annotations, DNA methylation data from 2,087 PD-associated DMCpGs and gene expression data from all 36,079 transcripts were overlapped. We detected 2,430 annotating pairs. Subsequently, Spearman correlation analysis was done under a FDR-adjusted $P < 0.05$.

**Transcriptional and epigenomic characterization of differentially methylated sites**

We analyzed our data in the context of functional chromatin states using the NHEK keratinocyte as background cell line (Ernst *et al*, 2011) which is similar to the cells used to generate iPSCs in our study. We considered regions with states 1 and 2 (designated as "active promoter" and "weak promoter") as "promoter regions"; state 3 as "poised promoters"; states 4, 5, 6, and 7 (designated as "strong enhancer" and "weak/poised enhancer") as "enhancer regions"; state 8 as "insulator"; states 9 and 10 (designated as "transcriptional transition" and "transcriptional elongation") as "strong transcription regions"; state 11 as "weak transcription"; state 12 as "Polycomb repressed"; and states 13, 14, and 15 (designated as "heterochromatin (nuclear lamina)" and "repetitive heterochromatin") as "heterochromatin regions".

**Transcription factor analysis**

We overlapped the methylation data from 2,087 PD DMCpGs with transcription factor (TF)-binding site (TFBS) clusters generated by ChIP-seq in the ENCODE project (Dunham *et al*, 2012; Gerstein *et al*, 2012; Lee *et al*, 2012) and available at the UCSC Genome Browser (http://genome.ucsc.edu/cgi-bin/hgTrackUi?db = hg19&g = wgEncodeHaibTfbs). For each TF, we calculated the $\log_2$ ratio of the proportion of TFBSs in the lists of hyper- and hypomethylated DMCpGs as compared to that in the whole methylation array (Table EV7). A Fisher exact test with a FDR-adjusted $P < 0.05$

resulted in 25 significant TFs. Of these, 23 TFs showed enrichment of binding sites at PD-hypermethylated DMCpGs. Spearman correlation analysis was done under a FDR $P < 0.05$ to study the association between average methylation levels at hypermethylated DMCpGs from enhancer sites, annotated in the NHEK background, and the gene expression of related TFs.

## Immunoblotting

We used the following primary antibodies: rabbit anti-FOXA1 (Abcam, 1:500, #ab23738), rabbit anti-NR3C1 (Santa Cruz, 1:2,000, #sc1002), goat anti-HNF4A (Santa Cruz, 1:1,000, #sc6556), rabbit anti-FOSL2 (Santa Cruz, 1:500, #sc604), rabbit anti-OTX2 (Millipore, 1:1,000, #ab9566), rabbit anti-PAX6 (Covance, 1:500, #prb278p), rabbit anti-ZIC1 (Abcam, 1:1,000, #ab72694), mouse anti-SYT11 (Santa Cruz, 1:500, #sc365991), rabbit anti-DCT (Santa Cruz, 1:250, #sc25544), mouse anti-DCC (BD Pharmigen, 1:2,000, #554223), mouse anti-SNCA (BD Transduction Laboratories, 1:500, #610787), mouse anti-NEFL (Sigma, 1:500, #5139), mouse anti-β-actin (Sigma, 1:5,000, #A5441), and mouse anti-α-tubulin (Sigma, 1:10,000, #T5168). Nitrocellulose membranes were incubated overnight at 4°C with primary antibodies diluted in 2% BSA/PBS. Appropriate secondary antibody (anti-mouse and anti-rabbit from Amersham Biosciences, or anti-goat from Santa Cruz), coupled to horseradish peroxidase and diluted in 1% milk powder/PBS (1:5,000 dilution), were incubated 1 h at room temperature, followed by repeated washing with PBS. Immunoreactive bands were visualized using the SuperSignal Femto Chemiluminescent Substrate (Pierce) in the ImageQuant RT ECL imaging system (GE Healthcare). Protein band intensity was quantified by densitometry using the ImageJ Software. A total of 10 PD samples and four controls were studied for OTX2, PAX6, ZIC1, SYT11, DCT, DCC, SNCA, and NEFL. In the case of FOXA1, NR3C1, HNF4A, and FOSL2, six PD cases and three controls were analyzed as to illustrate extremes of a gene expression continuum that gradually increases from PD to controls. In both cases, a two-tailed Student's *t*-test was used to compare the PD and control groups. All samples were assayed in at least three independent experiments. Data in Figs 7A and B, and EV3A and B are represented as group mean ± SEM.

## Immunofluorescence

iPSC-derived DAn were grown on plastic cover slide chambers, fixed with 4% paraformaldehyde, and then permeabilized with 0.5% Triton X-100 in TBS. Cells were then blocked in 0.5% Triton X-100 with 3% donkey serum for 2 h before 4°C overnight incubation with the appropriate primary antibodies. We used the following antibodies: mouse anti-TH (Chemicon, 1:1,000, #mab5280), rabbit anti-TH (Sigma, 1:1,000, #t8700), mouse anti-TUJ1 (Covance, 1:500, #mms435p), rabbit anti-GIRK2 (Sigma, 1:40, #P8122), goat anti-FOXA2 (R&D Systems, 1:100, #AF2400), rabbit anti-NURR1 (Santa Cruz, 1:200, #sc-991), mouse anti-SYP (Millipore, 1:500, #MAB332), rabbit anti-OTX2 (Millipore, 1:1,000, #ab9566), rabbit anti-PAX6 (Covance, 1:100, #prb278p), and mouse anti-DCC (BD Pharmigen, 1:250, #554223). Secondary antibodies used were all from the Alexa Fluor Series (Invitrogen, all 1:500). Images were taken using a Leica SP5 confocal microscope. For quantification of stained cells, randomly 300 cells per differentiated aggregate were counted

### The paper explained

**Problem**

As a complex multifactorial neurodegenerative disorder, pathogenic mechanisms of Parkinson's disease (PD) remain poorly understood. This is in part due to the inaccessibility to the dopaminergic neurons (DAn) targeted by disease which are only available postmortem.

**Results**

Upon cell reprogramming of somatic skin cells (fibroblasts) into induced pluripotent stem cells (iPSC), we generated DAn from patients with sporadic PD (sPD) (90-95% of cases) and patients with a less frequent familial form caused by mutations in the gene *LRRK2* (L2PD). Using genomewide approaches, we found large epigenomic (DNA methylation) and gene expression changes in DAn from PD patients as compared to healthy subjects. Interestingly, these changes were largely similar in sPD and familial L2PD. In addition, the PD epigenetic changes were specific for DAn since fibroblasts, iPSCs, or other neural types (not-DAn) derived from the same PD patients did not show epigenetic abnormalities. Moreover, the PD epigenetic changes affected prominently to regulatory regions (hypermethylation of enhancers) and were related to the gene or protein downregulation of a network of transcription factors which is relevant to PD.

**Impact**

Using a patient-specific iPSC-derived DAn model, our findings indicate that: (i) epigenetic deregulation occurs in PD, (ii) PD-associated changes are similar in the sporadic and the L2PD familial form suggesting common etiologic processes and potential applicability of common therapeutic treatments, and (iii) future iPSC-based cell therapy strategies should pay attention to the correct epigenomic status of the reprogrammed DAn if using patient own cells.

(average 5–6 differentiated aggregates per experiment). Data points represent the average of at least three independent experiments. To visualize nuclei, slides were stained with 0.5 µg/ml DAPI (4′,6-diamidino-2-phenylindole) and then mounted with PVA/DABCO.

## Study approval

This study has been conducted conforming the principles of the Declaration of Helsinki and the Belmont Report. The Commission on Guarantees for Donation and Use of Human Tissues and Cells of the Instituto de Salud Carlos III (ISCIII) and the local ethics committee at the Hospital Clínic de Barcelona approved the study. All subjects gave written informed consent prior to their participation in the study.

## Accession number

Genomewide DNA methylation and gene expression datasets generated in this study have been deposited in the Gene Expression Omnibus (GEO) under accession GSE16453. Summaries of analyses on these crude data may be found at Tables EV1, EV3, EV4, and EV6.

**Expanded View** for this article is available online.

## Acknowledgements

We are indebted to the patients with PD who have participated in this study and to their relatives. We are grateful to Prof. Carlos López-Otín and to Dr. Ariadna Laguna Tuset for critical reading of this manuscript. We thank Manel

Fernández and Cristina Muñoz for excellent technical assistance. We also acknowledge Carla Sureda and Ana Molgosa for helpful assistance with the neuronal differentiation of iPSCs. We are grateful to the Spanish National Genotyping Centre (CeGen, http://www.cegen.org) and to the Advanced Fluorescence Microscopy Unit of the Institut de Biologia Molecular de Barcelona (IBMB). Part of this work was developed at the Centre de Recerca Biomèdica Cellex and the Centre Esther Koplowitz, Barcelona, Spain. The study was funded by the Instituto de Salud Carlos III (ISCIII) through the Cooperative Projects program of the Centro de Investigación Biomédica en Red de Enfermedades Neurodegenerativas (CIBERNED) (to E.T., M.V., A.R., J.A. y J.L.-B.). Additional support was provided by the Spanish Ministry of Economy and Competitiveness (MINECO) grant FIS2010-21924-C02-02 and the Generalitat de Catalunya grant 2009SGR14 (to J.S.), the Fundación Botín (to J.L.-B.), the SAF program of the Spanish Ministry of Innovation and Science (to J.L.-B., J.M.C. and J.A.), the Spanish Cell Therapy Network (Red de Terapia Celular) of the ISICIII (to J.M.C.), the SAF2012-33526 grant and the Cell Therapy Network (TerCel nodes RD12/0019/0019,/0003, and/ 0033) of the ISCIII (to A.R.), the PIE14/00061 grant of the ISICIII/FEDER (to A.R., R.F.-S. and M.E.), the FIS project PI10/849 of the ISCIII (to M.V.), the BFU2013-49157-P grant, as well as the ERC-2012-StG grant of the European Research Council (ERC) (to A.C.), and the Mendelian Forms of Parkinson's Disease grant (MEFOPA) of the EC (to E.T.). R.F.-S. was supported by a Marie Skłodowska-Curie contract of the EC and IDIBAPS, and by a Juan de la Cierva contract of the Spanish Ministry of Economy and Competitiveness (MINECO), I.C.-C. by a CIBERNED contract, J.I.M.-S. by a Ramón y Cajal contract of the MINECO, and M.E. by a Miguel Servet contract of the ISCIII.

## Author contributions

RF-S and ME conceived the study. IC-C and GC contributed equally to this work. RF-S, AC, JIM-S, ME, and ET jointly supervised research. RF-S, IC-C, AR, MV, AC, JIM-S, and ME conceived and designed the experiments. RF-S, IC-C, RT, YR, AS-D, RV-B, JS, and ME performed the experiments. RF-S, IC-C, GC, AS-P, JLM, JIM-S, and ME performed statistical analyses. RF-S, IC-C, GC, JS, JL-B, JMC, JA, AR, MV, AC, JIM-S, ME, and ET analyzed the data. GC, AR, MV, AC, JIM-S, and ET contributed reagents, materials, or analysis tools. RF-S, IC-C, GC, AS-D, JL-B, JMC, JA, AR, MV, AC, JIM-S, ME, and ET wrote the paper.

## Conflict of interest

The authors declare that they have no conflict of interest.

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
