## [Review Process File · EMBO Molecular Medicine]

Aberrant epigenome in iPSC-derived dopaminergic neurons from Parkinson's disease patients

Rubén Fernández-Santiago, Iria Carballo-Carbajal, Giancarlo Castellano, Roger Torrent, Yvonne Richaud, Adriana Sánchez-Danés, Roser Vilarrasa-Blasi, Alex Sánchez-Pla, José Luis Mosquera, Jordi Soriano, José López-Barneo, Josep M. Canals, Jordi Alberch, Ángel Raya, Miquel Vila, Antonella Consiglio, José I. Martín-Subero, Mario Ezquerra, & Eduardo Tolosa

Corresponding author: Rubén Fernández-Santiago, Institut d'Investigacions Biomèdiques August Pi i Sunyer (IDIBAPS)

Review timeline:

Submission date:	15 May 2015
Editorial Decision:	06 July 2015
Revision received:	28 July 2015
Editorial Decision:	20 August 2015
Revision received:	24 September 2015
Accepted:	25 September 2015

Transaction Report:

Editor: Céline Carret

1st Editorial Decision

06 July 2015

Thank you for the submission of your manuscript to EMBO Molecular Medicine and your continued patience during the review process. I am really sorry for the delay in getting back to you. As mentioned before, we unfortunately were unable to obtain the report from referee #2 initially invited. However, luckily, the external advisor I contacted kindly accepted to provide a full review in less than a week, so you'll find below 2 sets of reviews.

You will see that while both referees find the paper potentially interesting, they are also both critical. Referee 1 is mainly concerned by technical issues that need to be clarified and thoroughly explained. Referee 3 is more positive but because the data could have a very big impact in the field, this referee requests further confirmation of the data using at least another PD mutant distinct from LRRK2 mutation.

I would be happy to consider a major revision of the work if you can address the issues that have been raised, experimentally when needed. Please note that it is EMBO Molecular Medicine policy to allow only a single round of revision and that, as acceptance or rejection of the manuscript will depend on another round of review, your responses should be as complete as possible.

EMBO Molecular Medicine has a "scooping protection" policy, whereby similar findings that are

published by others during review or revision are not a criterion for rejection. Should you decide to submit a revised version, I do ask that you get in touch after three months if you have not completed it, to update us on the status.

Please also contact us as soon as possible if similar work is published elsewhere. If other work is published we may not be able to extend the normal revision period beyond three months. If you however need a longer revision time for other reasons, please contact us.

Please read below for important editorial formatting.

I look forward to receiving your revised manuscript.

***** Reviewer's comments *****

Referee #1 (Remarks):

This study is potentially of interest because it is not well studied if there is any significant difference in the epigenomic landscape in DAN from healthy, sporadic, and familial PD. Strikingly, this study indicates that PD (both sporadic and LRRK2-associated PDs) iPSC-derived DAN have dramatically different DNA methylation patterns compared to control iPSC-derived DAN. If these results hold, it implies that PD iPSC-derived DAN maintain their intrinsic epigenomic status after the procedures of reprogramming and in vitro differentiation. Although this study is potentially important, there are numerous major caveats and concerns.

Major concerns:

1. It is well known that iPSC lines exhibit significant differences between individual lines from the same source cells. In particular, these fluctuations are even wider if viral reprogramming methods are used. Since the authors used only three factors in viral vectors, the resulting iPSC lines are expected to contain significant clonal variations.
2. In vitro differentiation of iPSCs into DAN is a highly stochastic and random process and its efficiency is also known to be significantly different among individual lines. Thus, it is likely that subsequent analyses are expected to suffer from significant variations and fluctuations, as shown in Fig. 1E. Since DAN is a minor population (<30%) and fluctuates, the remaining majority cells (70% or more) may contribute to further noise and may lead to flawed interpretation. In this regard, it is difficult to understand and/or interpret that both familial and sporadic PC iPSCs exhibit significantly different methylation patterns from those of control iPSCs without single exception. How can it be explained? What are the underlying mechanisms?
3. The PD pathogenesis is well known to be caused by both genetic and environmental factors. In contrast, the results of this study appear to indicate that the epigenetic regulation is similar in familial and sporadic PDs and determined by their genetic status without any input of environment. This notion is largely in conflict with the vast majority of previous studies.

Specific concerns:

1. The authors applied a retrovirus-based system instead of a non-integrated system to generate iPSC lines from diverse primary fibroblasts. It is well-known that viral-based methods cause clonal variations among iPSC lines derived from the same somatic cells. Was there no variation among iPSC lines derived by a retrovirus-based system?
2. The authors transduced the cells with 3 reprogramming factors including Oct4, Sox2, and Klf4 (without Myc), in contrast to the general reprogramming method using four factors. Is there any specific reason why they used 3 factors in this approach? What were the differences in reprogramming efficiency?
3. In Fig 2B, the authors found 1,261 and 2,512 DMCpGs from L2PD vs Control and sPD vs Control, respectively. Although at least 1,200 CpGs are differentially methylated in L2PD than sPD compared to control, they reported 0 DMCpGs from L2PD vs sPD. How could they explain this discrepancy?
4. To increase the differentiation efficiency of iPSCs into DAN cells, the authors transduced the iPSCs with Lmx1a using a lentiviral system. It is not easy to control the ectopic expression levels of delivered genes by a lentiviral system. Regarding this, did the authors excise the ectopic Lmx1a when

the DAN cells are enriched at 30 days differentiation? If not, is there any variation or difference among the DAN cells with differentially expressing *Lmx1a*?

5. The authors compared control iPSC with L2PD- and sPD-iPSCs in the whole experiments. It will be more informative if they include positive control using the normal human ES cells.

Referee #3 (Remarks):

The study by Fernandez-Santiago investigates DNA-methylation via bisulfite-conversion of DNA followed by microarray analysis in iPSC cell-derived dopaminergic neurons (DAN) from sporadic PD patients and from mutant *LRRK2* carriers. They find that DNA-methylation is significantly altered in sporadic and monogenetic PD DAN's but not in other type of neurons derived from the same fibroblasts or in undifferentiated cells. Most strikingly, there is no difference when comparing sporadic to monogenetic PD cases. Mechanistically the authors provide evidence that DAN in PD may fail to decrease DNA-methylation at selected loci during differentiation and thus show reduced expression of key transcription factors. This data is highly interesting and has a number of very important implications but raises also various questions.

1.

The most intriguing finding is the fact that control DAN'S differ from sporadic and genetic PD but that the direct comparison between sporadic and genetic cases did not yield significant differences. The authors thus combine both PD groups in their further analysis. However, this point is very critical and should be addressed in more detail. The data implies that most likely the genetic make up of the fibroblasts must drive DNA-methylation changes upon differentiation. With other words, even the sporadic cases investigated here are true genetic cases but potentially linked to a number of SNPs/CNVs in various genes that only have a limited impact alone but still act in the same pathway as *LRRK2* mutations. It is still hard to believe that all 4 sporadic cases by chance resemble the genetic deficits of *LRRK2* carriers. If so, this would be a major finding with enormous impact. Thus, to convincingly strengthen this very important finding it would be advised to test DAN's from individuals that carry at least one other mutation than *LRRK2*.

2.

The data shown in supplemental table 3 seems to be very crucial since it shows that 75% of the differentially methylated CpGs identified in PD (sporadic or mutant) do not change upon differentiation and thus implies that the changes seen in DAN'S are due to impaired differentiation potential. This should become a real figure and should also be performed individually for sporadic and mutant changes in DNA-methylation.

3.

1261 differentially methylated CpGs are detected in *LRRK2* DAN's and 2512 in sporadic PD. The authors state that 78% of the *LRRK2* changes are also seen in cells from sporadic DAN's and thus for further analysis sporadic and genetic PD forms are treated as one group. It would be still interesting to see to what genes/elements the 1251 regions specific to sporadic PD belong.

4.

The gene-expression data should be analyzed similar to DNA-methylation. Thus, how is the gene-expression in undifferentiated cells and in non-DAN's? Does the gene-expression in PD resembles too an undifferentiated state?

5.

A major finding of this study seems to be that even sporadic PD is a developmental disease and that DAN's acquire a specific deficit during differentiation. Is it thus not surprising that the observed changes are more severe in sporadic than in monogenetic PD? What is the authors explanation to this?

Article #EMM-2015-05439: Point-by-Point Response to Reviewers

We enclose herewith our manuscript '*Aberrant epigenome in iPSC-derived dopaminergic neurons from Parkinson disease patients*' (#EMM-2015-05439) to be further considered by EMBO Molecular Medicine.

We thank you and the Reviewers for the time dedicated to evaluate our work. We are very grateful for the constructive comments from the reviewers and for the encouragement to respond and resubmit a revised version of the manuscript. We herein enclose our Point-by-Point responses to all Reviewers' comments which overall have been highly helpful to implement our manuscript. All the points raised by the reviewers have been addressed in the revised manuscript and specific additional experimental work has been performed following Reviewers' advice.

One specific comment from Reviewer #3 requested to repeat the entire experiment by including new familial PD patients with mutations in other genes than *LRRK2*. Answering to this comment is out of scope of our work and would involve the recruitment of new patients, reprogramming of skin cells and epigenetic and transcriptomic characterization. These experiments would unnecessarily delay the publication of our findings for long time and thus, if possible, we would kindly ask the editor not to consider this specific comment for the overall re-evaluation of our work.

For the aforementioned reasons and given the merit of our responses, we hope that our edited manuscript would be ultimately considered for publication at EMBO Molecular Medicine.

Thanking you in advance for your consideration, and looking forward to hearing from your comments at the earliest of your convenience.

Sincerely

Rubén Fernandez-Santiago

Point-by-Point response to Referees

REFEREE #1

This study is potentially of interest because it is not well studied if there is any significant difference in the epigenomic landscape in DAN from healthy, sporadic, and familial PD. Strikingly, this study indicates that PD (both sporadic and LRRK2-associated PDs) iPSC-derived DAN have dramatically different DNA methylation patterns compared to control iPSC-derived DAN. If these results hold, it implies that PD iPSC-derived DAN maintain their intrinsic epigenomic status after the procedures of reprogramming and in vitro differentiation. Although this study is potentially important, there are numerous major caveats and concerns.

Major concerns:

Question 1.1. *It is well known that iPSC lines exhibit significant differences between individual lines from the same source cells. In particular, these fluctuations are even wider if viral reprogramming methods are used. Since the authors used only three factors in viral vectors, the resulting iPSC lines are expected to contain significant clonal variations.*

Response 1.1: We fully agree with the reviewer's concern. This is why we made every effort to ensure that the iPSC lines used here are comparable and representative of the disease condition. We used retroviral delivery of OCT4, KLF4 and SOX2 to generate 2–6 independent iPSC lines per individual, totalling 50 iPSC lines. Of those, 2 lines per patient were thoroughly characterized and shown to be fully reprogrammed to pluripotency, as judged by colony morphology and growth dynamics, sustained long-term passaging (>20 passages), karyotype stability, alkaline phosphatase (AP) staining, expression of pluripotency associated transcription factors (OCT4, SOX2, NANOG, CRIPTO and REX1) and surface markers (SSEA3, SSEA4, TRA1-60 and TRA1-81), silencing of retroviral transgenes, demethylation of OCT4 and NANOG promoters, in vitro pluripotent differentiation ability and generation of teratomas comprising derivatives of the three main embryo germ layers (Sánchez-Danés et al. 2012, Table 1 and Fig 1A–N). The efficiency of iPSC generation varied among different individuals, but did not depend on the presence or type of disease, nor on the age of donors. Whereas, most of the iPSC lines analyzed met our criteria for bona fide pluripotent stem cells, those that failed to silence the reprogramming transgenes, did not differentiate appropriately in vitro, or presented karyotype alterations were identified and excluded from further studies (Sánchez-Danés et al. 2012, Table 1). Overall, iPSC generated from PD patients or from healthy individuals were indistinguishable in all tests performed, with the exception that LRRK2-PD iPSC lines carried the *LRRK2* G2019S mutation (Sánchez-Danés et al. 2012, Fig 1O).

In summary, having several lines (between 2-6 lines per each patient), and several patients per disease condition (4 L2PD, 6 sPD) enabled us to test whether any difference could be disease-, patient-, or clone-specific (Sánchez-Danés et al. 2012). The fact that our current methylation analysis also clusters PD samples separate from controls provides further reassurance that the origin of the differences stems from the particular disease condition, rather than from any experimentally-induced clonal variation which on the other hand would be expected to occur randomly.

Sánchez-Danés A. et al. Disease-specific phenotypes in dopamine neurons from human iPS-based models of genetic and sporadic Parkinson's disease. *EMBO Mol Med.* 4(5):380-95 (2012).

Question 1.2. *In vitro* differentiation of iPSCs into DAn is a highly stoichiastic and random process and its efficiency is also known to be significantly different among individual lines. Thus, it is likely that subsequent analyses are expected to suffer from significant variations and fluctuations, as shown in Fig. 1E (?). Since DAn is a minor population (<30%) and fluctuates, the remaining majority cells (70% or more) may contribute to further noise and may lead to flawed interpretation.

Response 1.2. We agree with the reviewer that the accompanying not-DAn cells represent a source of noise. However, we should also note that this potential confounder has affected in similar manner to DAn cells from controls, L2PD patients and sPD patients since we have previously performed detailed analyses of the DAn differentiation ability of these iPSC lines and found no significant differences among the three groups (See Fig. 1E, Table 1, Reference Sanchez-Danes et al., 2012b, and response to major concern 1.1).

Yet we concur with the reviewer that it is formally possible that the differences we found between control and PD cultures could be due to the remaining 70% of cells that are not DAn. To address this indirectly, since having cultures consisting in 100% DAn is not technically feasible, we performed the reverse experiment and tested cultures with nearly 100% neurons, but no DAn (Figure 1D). Specifically, we differentiated iPSCs from the same PD patients and controls to generate iPSC-derived neural cultures not-enriched-in-DAn by omitting the lentiviral-mediated forced expression of the ventral midbrain determinant LMX1A as well as DAn patterning factors leading to a >4-fold depletion of the total final number of DAn (page 19, lines 309-405). In this case we could not find any DNA methylation differences between control and PD cultures, further indicating that identified PD-associated 2,087 DMCpGs were indeed attributable to the DAn component in our cultures.

Potentially confounding methylation signals from accompanying not-DAn cells, which would be expected to occur randomly, could have indeed contributed to an underestimation of methylation differences rather than to a flaw, a specific point in which we disagree with the reviewer.

Question 1.3. *In this regard, it is difficult to understand and/or interpret that both familial and sporadic PC iPSCs exhibit significantly different methylation pattern from those of control iPSCs without single exception. How can it be explained? What are the underlying mechanisms?*

Response 1.3. Although L2PD and sPD did epigenetically behave largely similar, the sPD patient PD-06 was an exception showing deviating epigenetic and transcriptomic profiles which were more similar to controls than to PD patients (See Figure 1D and Figure 2A). Interestingly this patient uniquely had a diagnosis of PD plus concomitant epilepsy (treated over the years with anti-epileptics), with a true non-pharmacogenic diagnosis of PD.

The epigenetic high resemblance between L2PD and sPD has been found using unbiased genome-wide methylation and expression analyses. In another independent study (under preparation), we have profiled the genome-wide microRNA expression of iPSC-DAn from the same patients and consistently found similar microRNA profiles for L2PD and sPD with PD-06 being also the exception which is more similar to controls. We agree with the reviewer that our findings are novel and remarkable. However, it should be taken into account that our results are largely in line with the unique clinical and neuropathological similarities of L2PD and sPD reported in the literature which is not observed for other monogenic forms of PD. Since we did not explore underlying mechanisms we still cannot establish the initiating cause of the identified epigenomic changes. Still following the reviewer's comment we have added a new section discussing the role of environmental and/or genetic factors potentially arising from

different but converging pathways which could ultimately cause common end-point alterations in L2PD and sPD (pages 12-13, lines 233-250 and 255-256).

Question 1.4. *The PD pathogenesis is well known to be caused by both genetic and environmental factors. In contrast, the results of this study appear to indicate that the epigenetic regulation is similar in familial and sporadic PDs and determined by their genetic status without any input of environment. This notion is largely conflict with the vast majority of previous studies.*

Response 1.4. To respond to this very constructive comment from the reviewer several considerations are to be taken into account. (i) The epigenome is expected to reflect the influence from the environment and accordingly we do not discard a common effect of unknown environmental factors in the epigenome from L2PD and sPD patients. Indeed, this would be in agreement with the environmental contribution for sPD and also with the environmental involvement explaining the reduced penetrance observed for *LRRK2* mutations which are expected to be modulated by environmental factors (Healy et al. 2008). (ii) The hypothesis that the genetic background of the PD fibroblasts could drive DNA methylation changes upon differentiation is a reasonable not only for L2PD but also for sPD. In fact, it has been recently described that sPD could be caused by an accumulation of common polygenic alleles with relatively low effect sizes (Escott-Price et al., 2015). (iii) Thus a possible integrative interpretation of our findings is that cumulative genetic risk factors such as single nucleotide polymorphisms (SNPs)/ copy number variants (CNVs) in sPD, or *LRRK2* mutations in L2PD, alone or in combination with still unknown environmental factors could be underlying the common epigenetic changes detected in both familial L2PD (n=4) and sporadic PD (n=6) cases from our study. These environmental and/or genetic factors, which probably arise from different but converging pathways, could cause common end-point alterations or basal damages in peripheral tissues such as fibroblast (not only in neurons) which fully manifest only upon reprogramming into iPSC-DAn. For the sake of clarity we have edited these interpretations in the main manuscript by adding a new section (pages 12-13, lines 233-250 and 255-256).

Specific concerns:

Specific concern 1.1. *The authors applied retrovirus-based system instead of non-integrated system to generate iPSC lines from diverse primary fibroblasts. It is well-known that viral-based methods cause clonal variations among iPSC lines derived from same somatic cells. Was there no variation among iPSC lines derived by retrovirus-based system?*

Specific response 1.1. (See Response to major concern 1.1). We fully agree with the reviewer's concern. This is why we made every effort to ensure that the iPSC lines used here are comparable and representative of the disease condition. We used retroviral delivery of OCT4, KLF4 and SOX2 to generate 2–6 independent iPSC lines per individual, totalling 50 iPSC lines. Of those, 2 lines per patient were thoroughly characterized and shown to be fully reprogrammed to pluripotency, as judged by colony morphology and growth dynamics, sustained long-term passaging (>20 passages), karyotype stability, alkaline phosphatase (AP) staining, expression of pluripotency associated transcription factors (OCT4, SOX2, NANOG, CRIPTO and REX1) and surface markers (SSEA3, SSEA4, TRA1-60 and TRA1-81), silencing of retroviral transgenes, demethylation of OCT4 and NANOG promoters, in vitro pluripotent differentiation ability and generation of teratomas comprising derivatives of the three main embryo germ layers (Sánchez-Danés et al. 2012, Table 1 and Fig 1A–N). The efficiency of iPSC generation varied among different individuals, but did not depend on the presence or type of

disease, nor on the age of donors. Whereas, most of the iPSC lines analyzed met our criteria for bona fide pluripotent stem cells, those that failed to silence the reprogramming transgenes, did not differentiate appropriately in vitro, or presented karyotype alterations were identified and excluded from further studies (Sánchez-Danés et al. 2012, Table 1). Overall, iPSC generated from PD patients or from healthy individuals were indistinguishable in all tests performed, with the exception that LRRK2-PD iPSC lines carried the *LRRK2* G2019S mutation (Sánchez-Danés et al. 2012, Fig 1O).

In summary, having several lines (between 2-6 lines per each patient), and several patients per disease condition (4 L2PD, 6 sPD) enabled us to test whether any difference could be disease-, patient-, or clone-specific (Sánchez-Danés et al. 2012). The fact that our current methylation analysis also clusters PD samples separate from controls provides further reassurance that the origin of the differences stems from the particular disease condition, rather than from any experimentally-induced clonal variation which on the other hand would be expected to occur randomly.

Sánchez-Danés A. et al. Disease-specific phenotypes in dopamine neurons from human iPSC-based models of genetic and sporadic Parkinson's disease. *EMBO Mol Med.* 4(5):380-95 (2012).

Specific concern 1.2. *The authors transduced the cells with 3 reprogramming factors including Oct4, Sox2, and Klf4 (without Myc), in contrast to the general reprogramming method using four factors. Is there any specific reason why they used 3 factors system in this approach? What was the differences in reprogramming efficiency?*

Specific response 1.2. Human somatic cells can be reprogrammed into iPSCs by the expression of a defined set of four factors (Oct4, Sox2, Klf4, and c-Myc) (Takahashi et al. 2006). However, it is possible to generate iPSCs from human fibroblasts in the absence of c-Myc retrovirus (Nakagawa et al. 2008, Wernig et al. 2008). Excluding c-Myc from the reprogramming cocktail is a safety measure since c-Myc is an oncogene that can induce tumorigenicity upon reactivation of the transgene, therefore hindering clinical applications. Based on our experience, although cells exposed to c-Myc are usually reprogrammed with higher efficiency by one to two orders of magnitude (also shown by Wernig et al. 2008), we have observed less clonal variation and tumor formation in cells obtained when c-Myc is not transduced. Reprogramming efficiencies using our method (without c-Myc) for selected clones included in this study are summarized in Figure 1E and Table 1.

Takahashi, K. & Yamanaka, S. Induction of pluripotent stem cells from mouse embryonic and adult fibroblast cultures by defined factors. *Cell* 126, 663–676 (2006).

Nakagawa, M. et al. Generation of induced pluripotent stem cells without Myc from mouse and human fibroblasts. *Nature Biotechnol.* 26, 101–106 (2008).

Wernig, M., Meissner, A., Cassady, J. P. & Jaenisch, R. c-Myc is dispensable for direct reprogramming of mouse fibroblasts. *Cell Stem Cells* 2, 11–12 (2008).

Specific concern 1.3. *In Fig 2B, the authors found 1,261 and 2,512 DMCPGs from L2PD vs Control and sPD vs Control, respectively. Although at least 1,200 CpGs are differentially methylated in L2PD than sPD compared to control, they reported 0 DMCPGs from L2PD vs sPD. How could they explain this discrepancy?*

Specific response 1.3. We found more DMCPGs in sPD than in L2PD using the same stringent significance cut-off which is requested for our methylation platform (delta-beta methylation difference above |0.25| and false discovery rate (FDR) adjusted P Wilcoxon rank test below 0.05) (Bibikova et al. 2011). The identification of more DMCPGs in sPD than in L2PD under the

same cut-off could be a consequence of the different number of studied subjects, i.e. n=6 in sPD and n=4 in L2PD and to the associated greater statistical power to detect differences in sPD. In addition, these numbers of DMCPGs do not necessarily imply a discrepancy of 1,251 CpGs or so between L2PD and sPD as suggested by the reviewer. These 'missing' differences expected by the reviewer would encompass CpGs in L2PD which did not pass the restrictive significance cut-off although the degree of methylation of these CpGs is more similar to sPD than to controls.

Specific concern 1.4. *To increase the differentiation efficiency of iPSCs into DAn cells, the authors transduced the iPSCs with Lmx1a using lentiviral system. It is not easy to control the ectopic expression levels of delivered genes by lentiviral system. Regarding this, did the authors excise the ectopic Lmx1a when the DAn cells are enriched at 30 days differentiation? If not, is there any variation or difference among the DAn cells with differentially expressing Lmx1a?*

Specific response 1.4. Lmx1a expression derived from the lentiviral vector should be tightly restricted to Nestin-expressing neural progenitor cells. Mature and post-mitotic neurons should not express Nestin. Moreover, in the article from Sanchez-Danes, et al. 2012 which is the methodological precedent of our current work it is shown that there are no significant differences in the efficiency of lentiviral transduction with Lmx1a in the Nestin-expressing neural precursors cells among individuals, and even it is not dependent on the presence or type of PD. However, we did not test whether there it could be some leaky ectopic expression of the vector, but we state that epigenomic differences are more penetrant than other biases.

Sanchez-Danes A, et al. Efficient generation of A9 midbrain dopaminergic neurons by lentiviral delivery of LMX1A in human embryonic stem cells and induced pluripotent stem cells. *Hum Gene Ther.* 23: 56-69 (2012).

Specific concern 1.5. *The authors compared control iPSC with L2PD- and sPD-iPSCs in the whole experiments. It will be more informative if they include positive control using the normal human ES cells.*

Specific concern 1.5. (For reviewer's perusal only). The importance of using iPSCs in our experimental design is that they are derived from the fibroblasts of the same donors/patients in which we later generated DAn. We agree that it may be meaningful to use ESCs as control, but we hope that the reviewer agrees with us that using the iPSCs from the very same donors/patients is probably a better control for the DAn samples in terms of experimental design. However, to properly answer this question and make sure that by using iPSCs we do not introduce a significant bias in our analyses, we have used 450k data from a published article (Nazor et al., 2012, see the full reference below) and have studied whether they look similar or not to our iPSCs. Indeed, as you can see in the figure below (Figure A) using the differentially methylated sites between DAn in controls and PD (main finding of our study, n=2,087 DMCPGs), the methylation of iPSCs and ESCs is largely similar.

Additionally, by performing an unsupervised analysis using all the CpG methylation values measured in the array, we also observe that iPSCs and ESCs cluster close to each other (see figure B below). There are certainly some differences, as shown previously in other articles, but this finding globally reflects that the epigenetic reprogramming of fibroblasts into iPSCs in our manuscript achieves a DNA methylation signature similar to ESCs.

Nazor KL et al. Recurrent variations in DNA methylation in human pluripotent stem cells and their differentiated derivatives. *Cell Stem Cell.* 10(5):620-634 (2012).

New Figure to Answer Specific Concern 1.5

A

B

REFEREE #3:

The study by Fernandez-Santiago investigates DNA-methylation via bisulfite-conversion of DNA followed by microarray analysis in iPS cell-derived dopaminergic neurons (DAn) from sporadic PD patients and from mutant LRRK2 carriers. They find that DNA-methylation is significantly altered in sporadic and monogenetic PD DAn's but not in other type of neurons derived from the same fibroblasts or in undifferentiated cells. Most strikingly, there is no difference when comparing sporadic to monogenetic PD cases. Mechanistically the authors provide evidence that DAn in PD may fail to decrease DNA-methylation at selected loci during differentiation and thus show reduced expression of key transcription factors. This data is highly interesting and has a number of very important implications but raises also various questions.

Question 3.1. *The most intriguing finding is the fact that control DAn'S differ from sporadic and genetic PD but that the direct comparison between sporadic and genetic cases did not yield significant differences. The authors thus combine both PD groups in their further analysis. However, this point is very critical and should be addressed in more detail. The data implies that most likely the genetic make-up of the fibroblasts must drive DNA-methylation changes upon differentiation. With other words, even the sporadic cases investigated here are true genetic cases but potentially linked to a number of SNPs/CNVs in various genes that only have a limited impact alone but still act in the same pathway as LRRK2 mutations. It is still hard to believe that all 4 sporadic cases by chance resemble the genetic deficits of LRRK2 carriers. If so, this would be a major finding with enormous impact. Thus, to convincingly strengthen this very important finding it would be advised to test DAn's from individuals that carry at least one other mutation than LRRK2.*

Response 3.1. To respond to this very constructive comment from the reviewer several considerations are to be taken into account. (i) The epigenome is expected to reflect the influence from the environment and accordingly we do not discard a common effect of unknown environmental factors in the epigenome from L2PD and sPD patients. Indeed, this would be in agreement with the environmental contribution for sPD and also with the environmental involvement explaining the reduced penetrance observed for *LRRK2* mutations which are expected to be modulated by environmental factors (Healy et al. 2008). (ii) The hypothesis that the genetic background of the PD fibroblasts could drive DNA methylation changes upon differentiation is a reasonable not only for L2PD but also for sPD. In fact, it has been recently described that sPD could be caused by an accumulation of common polygenic alleles with relatively low effect sizes (Escott-Price et al., 2015). (iii) Thus a possible integrative interpretation of our findings is that cumulative genetic risk factors such as single nucleotide polymorphisms (SNPs)/ copy number variants (CNVs) in sPD, or *LRRK2* mutations in L2PD, alone or in combination with still unknown environmental factors could be underlying the common epigenetic changes detected in both familial L2PD (n=4) and sporadic PD (n=6) cases from our study. These environmental and/or genetic factors, which probably arise from different but converging pathways, could cause common end-point alterations or basal damages in peripheral tissues such as fibroblast (not only in neurons) which fully manifest only upon reprogramming into iPSC-DAn. For the sake of clarity we have edited these interpretations in the main manuscript by adding a new section (pages 12-13, lines 233-250 and 255-256).

We agree with the reviewer that it would be interesting to test DAn from individuals that carrying other mutations in other PD genes. However, studying other monogenic forms of PD is currently out of the scope of our study and would delay for long time the publication of our timely findings. It should be also mentioned that we have specifically focussed in L2PD since this is the only monogenic form which really resembles the common sPD form (90-95% of cases). Other monogenic forms encompass rare autosomal-recessive early-onset forms (Parkin, PINK1, DJ-1)

which clinically or neuropathological deviate from sPD, or which are extremely unfrequent (α -synuclein point mutations and gene duplications).

Question 3.2. *The data shown in supplemental table 3 seems to be very crucial since it shows that 75% of the differentially methylated CpGs identified in PD (sporadic or mutant) do not change upon differentiation and thus implies that the changes seen in DAN'S are due to impaired differentiation potential. This should become a real figure and should also be performed individually for sporadic and mutant changes in DNA-methylation.*

Response 3.2. Following the reviewer suggestion we have constructed a new figure (Supplemental Figure 2) representing data included in Supplementary Table 3 and showing that 75% of all the 2,087 PD-associated DMCpGs did not change in the differentiation from iPSCs to DAN in PD patients (right), whereas only 40% DMCpGs remained unchanged in controls (left). This figure illustrates the incomplete epigenomic remodeling in PD in spite of the successful reprogramming and differentiation into mature DAN.

New Supplemental Figure 2

Supplementary Figure 2. Differentially methylated CpGs (DMCpGs) detected in iPSC-derived DAN from PD patients (n=2,087) and comparison of their methylation status with respect to all undifferentiated iPSCs. DNA methylation data derived from main Figure 2D reorganized to show that iPSC-derived DAN from PD patients show a total of 75% of all the 2,087 PD-associated DMCpGs did not change in the differentiation from iPSCs to DAN in PD patients (right), whereas only 40% DMCpGs remained unchanged in controls (left), suggesting an incomplete epigenomic remodeling in PD in spite of the successful reprogramming and differentiation into mature DAN.

Question 3.3. 1261 differentially methylated CpGs are detected in LRKK2 DAN's and 2512 in sporadic PD. The authors state that 78% of the LRRK2 changes are also seen in cells from sporadic DAN's and thus for further analysis sporadic and genetic PD forms are treated as one group. It would be still interesting to see to what genes/elements the 1251 regions specific to sporadic PD belong.

Response 3.3. To answer the inquiry for the reviewer we have disclosed the specific DMCpGs and corresponding annotated genes from L2PD which are common to sPD (78% of the 1,261 DMCpGs). Please see Supplementary Table 1, sheet 2 ('L2PD vs Controls') in which we have specifically depicted in bold lettering the DMCpGs and annotated genes from L2PD which are common to sPD (78%) whereas DMCpGs specific for L2PD alone appear in normal letters.

Question 3.4. The gene-expression data should be analyzed similar to DNA-methylation. Thus, how is the gene-expression in undifferentiated cells and in non-DAN's? Does the gene-expression in PD resembles too an undifferentiated state?

Response 3.4. We have taken into account the comment from the reviewer and performed the following experiment. We selected the top 10 differentially expressed genes (DEGs) detected in PD iPSC-derived DAN and studied their expression levels in iPSC-derived neural cultures not-enriched-in-DAN (n=6 PD, n=3 controls) and in undifferentiated iPSC (n=4 PDs, n=3 controls). Studied genes included OTX2, PAX6, ZIC1, DCT, NEFL, DCC, SYT11, SNCA, CAV1 and TFP12. The relative log2 fold change (FC) values and FDR-adjusted significance P-values in PD patients vs. controls were calculated as described in the methods section (page 22, lines 473-486). Overall, we did not detect gene expression differences for any of the genes, neither in iPSC-derived neural cultures not-enriched-in-DAN nor undifferentiated iPSCs. Results from this new experiment are summarized in the table below and could be added to the manuscript.

iPSC-derived neural cultures not-enriched-in-DAN (PD vs Control)			Undifferentiated iPSC (PD vs Control)		
Gene	Fold-change	FDR P-value	Gene	Fold-change	FDR P-value
OTX2	1,424	0,733	OTX2	0,380	0,957
PAX6	1,167	0,795	PAX6	0,957	0,957
ZIC1	1,497	0,667	ZIC1	0,871	0,957
DCT	1,474	0,696	DCT	1,074	0,957
NEFL	1,588	0,667	NEFL	0,656	0,957
DCC	2,116	0,667	DCC	0,520	0,957
SYT11	2,517	0,667	SYT11	0,793	0,957
SNCA	1,933	0,667	SNCA	0,732	0,957
CAV1	1,403	0,667	CAV1	0,531	0,957
TFP12	1,539	0,667	TFP12	0,728	0,957

Question 3.5. *A major finding of this study seems to be that even sporadic PD is a developmental disease and that DAN's acquire a specific deficit during differentiation. Is it thus not surprising that the observed changes are more severe in sporadic than in monogenetic PD? What is the authors explanation to this?*

Response 3.5. Our conclusion stated in the article is that monogenic L2PD and the sPD share similar DNA methylation changes. The fact that we found more DMCPGs in sPD than in L2PD under the same stringent cut-off (as requested for our methylation platform, delta-beta methylation difference above |0.25| and a false discovery rate (FDR) adjusted *P* Wilcoxon rank test below 0.05) (Bibikova et al. 2011) could be a consequence of the different number of studied subjects, i.e. n=6 in sPD and n=4 in L2PD respectively and also to the associated greater statistical power in sPD to detect differences. In addition, the Principal Component Analysis (PCA) from Figure 2A evidences that sPD and L2PD are clustered and mixed altogether, and both separated from controls, and the same applies to the heatmap shown in Figure 2D. Thus, our presented data are not conclusive enough to affirm that changes are more severe in sporadic than in monogenetic PD. For this reason we have concluded that methylation changes in L2PD and sPD appear to be similar.

2nd Editorial Decision

20 August 2015

Thank you for the submission of your revised manuscript to EMBO Molecular Medicine. We have now received the enclosed report from the referee who was asked to re-assess it. As you will see, this reviewer is now supportive and I am pleased to inform you that we will be able to accept your manuscript pending final amendments:

1) Please make sure to discuss adequately the limitations of the conclusions raised as only one genetic PD model was used, as commented by referee 3. Please provide a letter INCLUDING the reviewer's reports and your detailed responses to their comments (as Word file).

Please submit your revised manuscript within two weeks. I look forward to seeing a revised form of your manuscript as soon as possible.

***** Reviewer's comments *****

Referee #3 (Comments on Novelty/Model System):

Still a very interesting finding that will cause some controversy in the field. This impact will be high. Using iPS cells is generally problematic but adequate.

Referee #3 (Remarks):

I do not completely follow the authors response to question 3.1. Their response is basically repeating the question with different words but without other data on genetic PD cases there will be no conclusive answer. Again, in my view the findings presented here - if confirmed - will have major implications that will go beyond the authors current interpretation.

I do see the authors point that it will be major work to repeat the experiments in other genetic forms of PD.

I suggest to publish as it is.

2nd Revision - authors' response

24 September 2015

REFEREE #3

(Comments on Novelty/ Model System): *Still a very interesting finding that will cause some controversy in the field. This impact will be high. Using iPS cells is generally problematic but adequate.*

(Remarks): *I do not completely follow the authors response to question 3.1. Their response is basically repeating the question with different words but without other data on genetic PD cases there will be no conclusive answer. Again, in my view the findings presented here - if confirmed - will have major implications that will go beyond the authors current interpretation. I do see the authors point that it will be major work to repeat the experiments in other genetic forms of PD. I suggest to publish as it is.*

Response: We agree with the reviewer's comment and have acknowledged it the edited manuscript as it follows. Line 229: "However, one limitation of our work is that we studied only one monogenic form of PD (L2PD) and therefore the epigenetic involvement in other PD genetic forms remains to be explored in future studies".